# DPAIL: Training Diffusion Policy for Adversarial Imitation Learning without Policy Optimization

**Yunseon Choi**[1]    **Minchan Jeong**[1]    **Soobin Um**[1,2]    **Kee-Eung Kim**[1]

[1]Kim Jaechul Graduate School of AI, KAIST
[2]Department of AI, Kookmin University
{cys9506, kekim}@kaist.ac.kr

## Abstract

Human experts employ diverse strategies to complete a task, producing to multi-modal demonstration data. Although traditional Adversarial Imitation Learning (AIL) methods have achieved notable success, they often collapse theses multi-modal behaviors into a single strategy, failing to replicate expert behaviors. To overcome this limitation, we propose **DPAIL**, an adversarial IL framework that leverages diffusion models as a policy class to enhance expressiveness. Building on the Adversarial Soft Advantage Fitting (ASAF) framework, which removes the need for policy optimization steps, DPAIL trains a diffusion policy using a binary cross-entropy objective to distinguish expert trajectories from generated ones. To enable optimization of the diffusion policy, we introduce a novel, tractable lower bound on the policy's likelihood. Through comprehensive quantitative and qualitative evaluations against various baselines, we demonstrate that our method not only captures diverse behaviors but also remains robust as the number of behavior modes increases.

## 1 Introduction

Many sequential decision-making tasks involve multiple distinct expert behaviors to achieve the same goal, resulting in multi-modal demonstration data. For example, human demonstrations of robotic manipulation tasks often exhibit various grasping strategies, alternating between cautious and aggressive movements. Capturing such diversity is important in imitation learning (IL), as an agent should reproduce the range of expert strategies rather than collapse to an average behavior. Generative modeling approaches offer a potential solution by learning rich distributions over expert behavior. In particular, diffusion models, such as the Denoising Diffusion Probabilistic Model (DDPM) [15], have emerged as powerful generative models capable of capturing complex, multi-modal data distributions. Recent work [22] suggests that diffusion-based policies, trained to mimic expert actions via behavioral cloning, can more faithfully reproduce stochastic and multi-modal human demonstrations than conventional uni-modal policies. These successes motivate us to leverage diffusion models for IL, aiming to model diverse expert behaviors as a multi-modal distribution. However, a behavior cloned policy often struggles to generalize beyond the demonstrated data distribution, suffering from compounding errors when it encounters novel states [25].

While behavioral cloning directly learns a policy from state-action examples via supervised learning, inverse reinforcement learning (IRL) methods instead infer a reward function under which the expert is optimal and then train a policy to maximize this reward [1, 37]. By capturing the underlying objectives of expert behavior rather than specific actions, policies trained via IRL exhibit better generalization and adaptability to variations in the environment or task dynamics. However, classical IRL was often impractical, requiring repeatedly solving a full RL problem as an inner loop. Adversarial Imitation

Learning (AIL) reframes IRL as a distribution-matching problem, enabling more efficient learning. In particular, Generative Adversarial Imitation Learning (GAIL) [14] introduced a GAN-like framework in which a discriminator is trained to distinguish expert from agent behaviors, and a policy modeled as a generator is trained via policy gradient to produce trajectories that the discriminator cannot tell apart from expert trajectories. This formulation effectively matches the occupancy measure of the learned policy to that of the expert, and GAIL and its variants [36, 20] have demonstrated strong performance on complex control tasks. However, the alternating optimization between the discriminator and policy makes such adversarial methods computationally expensive and sometimes unstable.

Recent research has sought to simplify AIL by removing the costly policy optimization inner loop. Barde et al. [6] proposed Adversarial Soft Advantage Fitting (ASAF), which avoids policy gradient updates altogether. In ASAF, the discriminator is structured to output a parametric policy distribution directly, conditioned on both the previous generator policy and a new learnable policy. When optimized to equilibrium, this discriminator effectively solves for the optimal imitation policy, eliminating the need for a separate policy update step, i.e. the updated policy is obtained for free from the discriminator parameters. This elegant one-step formulation drastically simplifies training, cutting out the policy gradient updates. Crucially, however, ASAF assumes the policy class is one for which probabilities can be evaluated in closed form, e.g. policies modeled as Gaussian or normalizing flow. This is because the training objective of the discriminator involves computing policy densitites, an operation that is tractable for simple parametric or flow-based policies. This assumption breaks down for diffusion models: diffusion policies generate samples via an iterative denoising process, and their exact likelihood are generally intractable to compute. This incompatibility prevents a naive approach to ASAF to diffusion-based policies.

In this paper, we bridge diffusion generative models with AIL without requiring any policy optimization step. Building on the insights of ASAF, we introduce a diffusion policy imitation framework that retains the ability to match expert distributions adversarially while harnessing the expressive power of diffusion models for multi-modal behavior. The key contribution is a novel training objective: we derive a tractable lower bound to the ASAF objective that enables us to train diffusion policies efficiently despite the intractability of their exact densities. By optimizing this lower bound, our method avoids explicit computation of marginal diffusion densities, yet still directs the diffusion policy to fit expert demonstration distribution. Notably, our approach also complements recent advances that integrate diffusion models into AIL. For example, diffusion models have been used to strengthen the discriminator in AIL by modeling state-action distributions [18, 34]; in contrast, our focus in on using diffusion as the policy model itself and devising a training algorithm that circumvents policy gradient updates.

We summarize our main contributions as follows:

1. **Diffusion Policies for AIL:** To the best of our knowledge, we propose the first AIL framework that employs diffusion policies to model a multi-modal expert behavior distributions.

2. **ASAF for Diffusion Policies**: We adapt the ASAF algorithm [6] to diffusion-based policies by deriving a new training objective that bypasses intractable computation of marginal diffusion densities. We show that this surrogate objective is a tight lower bound of the training objective in ASAF, making our algorithm an instance of minorization-maximization algorithm [16].

3. **Empirical Validation:** We evaluate the proposed diffusion policy imitation method on standard IL benchmarks, including continuous control tasks in the MuJoCo simulator [32] and maze navigation tasks in the Maze2D environment [11]. Without the policy optimization loop, our diffusion-based approach effectively learns to imitate the expert demonstrations on these tasks, demonstrating that diffusion policies can capture the breadth of expert behaviors while maintaining competitive performance with state-of-the-art imitation learning methods.

## 2 Related Works

### 2.1 Adversarial Imitation Learning (AIL)

AIL recasts imitation learning as occupancy-measure distribution matching, removing the expensive full RL inner loop required by classical IRL algorithms. GAIL [14] was one of the first to formulate this idea as a GAN-style minimax game between a policy generator and a discriminator, proving its effectiveness in high-dimensional continuous-control tasks. A series of extensions broadened its

expressiveness: InfoGAIL [20, 13] augments the adversarial objective with a mutual information regularizer that pushes the policy to embed diverse behavioral patterns in latent variables, enabling it to disentangle and reproduce multiple expert behavior modes. Building on this idea, Ess-InfoGAIL [10] introduces a semi-supervised learning approach that extracts more meaningful representations from a small amount of labeled demonstrations, thereby improving sample efficiency and stability. Although these variants enhance the ability to imitate diverse expert behaviors, they still depend on the policy update in the inner loop, inheriting the associated computational complexity and instability from alternating optimization.

To alleviate this burden, Adversarial Soft Advantage Fitting (ASAF) [6] reformulates the AIL objective so that the policy update occurs implicitly inside the discriminator, eliminating the need for separate policy optimization. While computationally attractive, ASAF has so far been restricted to policy classes with probabilities evaluatable in closed form, e.g. Gaussians or normalizing flows, because its objective requires exact density evaluation.

## 2.2 Diffusion Models in Sequential Decision Making

Diffusion models [15, 30, 28, 31] have demonstrated remarkable success in generative modeling, particularly in the domains of images and audio [8, 23, 5], and are now increasingly being applied to sequential decision-making tasks. Diffusion-BC [22] showed that diffusion-based policies are more accurate in replicating diverse human demonstrations in robotic manipulation tasks under behavior cloning. Diffuser [17] uses diffusion model to generate trajectories guided by the reward function. Diffuser Decision [2] uses conditional diffusion model with classifier-free guidance to generate trajectories. Diffusion-QL [35] represents the policy as a diffusion model, and train it via Q-learning.

Diffusion models have been also adopted in the AIL setting. DiffAIL [34] replaces the standard binary discriminator in AIL with an unconditional diffusion model trained via diffusion loss, enabling more precise occupancy-measure matching of expert state–action pairs. DRAIL [18] instead incorporates a conditional diffusion model as the discriminator to improve its ability to distinguish between agent and expert state-action pairs.

Our work departs from these prior works by extending ASAF to diffusion-based policies, whose exact densities are intractable to evaluate, via a tractable lower bound on the ASAF objective. This development combines the efficiency and stability of ASAF with the rich, multi-modal expressivity of diffusion models, achieving adversarial imitation learning without any explicit policy optimization loop.

# 3 Preliminaries

## 3.1 Reinforcement Learning and Diffusion Models

**Markov Decision Process (MDP)**  We model the RL problem as an MDP, defined by the tuple $\mathcal{M} = (\mathcal{S}, \mathcal{A}, P, r, P_0, \gamma)$, where $\mathcal{S}$ is the state space, $\mathcal{A}$ is the action space, $P(s'|s, a) \in [0, 1]$ is the state transition probability function, $r(s, a) \in \mathbb{R}$ is the reward function, $P_0(s) \in [0, 1]$ is the initial state distribution, and $\gamma \in [0, 1]$ is a discount factor. The objective is to learn a policy that maximizes the expected return, $\max \mathbb{E}[\sum_t \gamma^t r(s_t, a_t)]$.

**Diffusion Models**  Diffusion models [15] have emerged as powerful tools for RL tasks [17, 19, 3]. Following [17], we utilize diffusion probabilistic models [15] for generating trajectories. These models are composed of two stochastic processes: forward noising process and reverse denoising process. The forward process, shown in Eq. (1), is a Markov chain with a predefined Gaussian transition, where data $\tau \in \mathcal{T}$ is iteratively corrupted according to a fixed variance schedule:

$$q(\tau^i|\tau^{i-1}) := \mathcal{N}(\tau^i; \sqrt{1 - \beta_i}\tau^{i-1}, \beta_i\mathbf{I}), \tag{1}$$

for $i \in \{1, \ldots, N\}$ and $0 < \beta_i < 1$. The variance schedule $\beta_i$ is chosen such that $q(\tau^N)$ approximates $\mathcal{N}(\mathbf{0}, \mathbf{I})$. The reverse (denoising) process, shown in Eq. (2), is another Markov chain with a *parameterized* Gaussian transition, used for the data sampling distribution $\tau^0 \sim p_\theta(\cdot)$:

$$p_\theta(\tau^{i-1}|\tau^i) := \mathcal{N}(\tau^{i-1}; \mu_\theta(\tau^i, i), \mathbf{\Sigma}^i) \tag{2}$$

where $\mu_\theta(\tau^i, i)$ is typically parameterized by a neural network. A common way to train this reverse process is via the variational lower bound. However, rather than predicting the mean $\mu$, DDPM [15] propose to predict the forward noise $\epsilon$, leading to the simplified loss function:

$$\mathbb{E}_{\tau^0, \epsilon \sim \mathcal{N}(\mathbf{0}, \mathbf{I}), i \sim \mathcal{U}(1, N)}[\|\epsilon - \epsilon_\theta(\tau^i, i)\|^2], \tag{3}$$

where $\tau^i := \sqrt{\alpha_i}\tau^0 + \sqrt{1 - \alpha_i}\epsilon$, and $\alpha_i := \prod_{j=1}^{i}(1 - \beta_j)$. Here, $\epsilon_\theta$ is the noise prediction network that estimates the noise $\epsilon$ added to the clean sample $\tau^0$ that leads to the noisy sample $\tau^i$.

### 3.2 Adversarial Imitation Learning

**GAIL [14]** This algorithm employs a GAN-like training for AIL, where a policy $\pi$ acts as a generator and an explicit discriminator $D : \mathcal{T} \rightarrow [0, 1]$ serves as a binary classifier to distinguish the trajectory $\tau \in \mathcal{T}$ between the demonstration distribution $p_E$ induced by the expert policy $\pi_E$ and the imitation policy distribution $p_\pi$. The policy and the discriminator are trained with the minimax optimization objective:

$$\min_\pi \max_D \mathcal{L}(D, p_\pi) \tag{4}$$
$$\text{where } \mathcal{L}(D, p_\pi) := \mathbb{E}_{\tau \sim p_E}[\log D(\tau)] + \mathbb{E}_{\tau \sim p_\pi}[\log(1 - D(\tau)].$$

GAIL alternates between updating discriminator and the policy. The discriminator is updated using the binary cross-entropy loss to differentiate expert samples from generated ones. The policy is then updated through RL by using the discriminator output as the reward, e.g. $-\mathbb{E}_{\tau \sim p_\pi}[\log(1 - D(\tau))]$.

**ASAF [6]** This algorithm, while adopting the same GAIL objective in Eq. (4), introduces the *structured discriminator*, motivated by the analytical solution of the inner maximization [9]. Formally, for any pair of policies $\pi$ and $\pi'$, define

$$D_{\pi, \pi'}(\tau) = \frac{p_{\pi'}(\tau)}{p_{\pi'}(\tau) + p_\pi(\tau)}. \tag{5}$$

Then, the solution to the inner maximization in Eq. (4) is equal to the expert distribution, $\pi^* = \text{argmax}_{\pi'} \mathcal{L}(D_{\pi, \pi'}, \pi) = \pi_E$ for any policy $\pi$, and the solution to the outer minimization is also equal to the same solution, i.e. $\text{argmin}_\pi \max_{\pi'} \mathcal{L}(D_{\pi, \pi'}, \pi) = \text{argmin}_\pi \mathcal{L}(D_{\pi, \pi_E}, \pi) = \pi_E$.

Based on this result, a practical implementation of ASAF sets $\pi$ as the policy from the previous iteration and updates the policy $\pi'$ to maximize the discriminator objective $\mathcal{L}$.

## 4 Method

Our goal is to employ a diffusion-based policy to capture the multi-modal behaviors observed in expert demonstrations. AIL methods typically involve training a discriminator and a generator via a minimax objective, which often leads to training instability. In particular, when the discriminator becomes too accurate, it can cause vanishing gradients for the generator, hindering the learning process. When applying such adversarial training to diffusion policies, these issues are further exacerbated. Specifically, computing policy gradient from reward signals requires backpropagation through the entire diffusion sampling steps, resulting in substantial computational cost and additional instability [21].

Therefore, we propose to train the diffusion policy using ASAF for improved efficiency and stability, as ASAF does not require an explicit discriminator or separate policy gradient steps. The fundamental bottleneck, however, is that ASAF requires efficient evaluations of $p_\pi(\tau)$, which is intractable for diffusion models. To address this issue, we derive a lower bound of the discriminator training objective that is tight at the optimal solution. With this lower bound as the surrogate objective, our algorithm, *Diffusion Policy Adversarial Imitation Learning* (**DPAIL**), becomes an instance of a Minorization-Maximization (MM) algorithm [16].

**DPAIL: Diffusion Policy for Adversarial Imitation Learning without Policy Optimization** In line with Janner et al. [17], we assume trajectory-level diffusion models $p_\theta(\tau)$, instead of single-step models. Throughout this paper, we denote $\tau^0$ as a trajectory to reuse notations from diffusion models.

First, we focus on the first term of Eq. (4) that involves expert demonstrations. Using the structured discriminator in Eq. (5), the maximization of the discriminator can be formulated as:

$$\max_{\theta} \mathbb{E}_{\tau^0 \sim p_E} \left[ \log D_{\theta^{\text{old}},\theta}(\tau^0) \right] = \max_{\theta} \mathbb{E}_{\tau^0 \sim p_E} \left[ \log \sigma \left( \log \frac{p_\theta(\tau^0)}{p_{\theta^{\text{old}}}(\tau^0)} \right) \right]. \tag{6}$$

When $p_\theta(\tau^0)$ and $p_{\theta^{\text{old}}}(\tau^0)$ are modeled as diffusion processes, it is computationally expensive to directly calculate the marginal densities. We first leverage the fact that the perturbed transitions $q(\tau^i|\tau^{i-1})$ of $p_\theta$ and $p_{\theta^{\text{old}}}$ are the same for all forward diffusion steps $i \in [1, N]$ since they have the same variance schedule. This allows us to rewrite the log density ratio as follows:

$$\log \frac{p_\theta(\tau^0)}{p_{\theta^{\text{old}}}(\tau^0)} = \log \frac{\prod_{i=1}^{N} q(\tau^i|\tau^{i-1})}{\prod_{i=1}^{N} q(\tau^i|\tau^{i-1})} \frac{p_\theta(\tau^0)}{p_{\theta^{\text{old}}}(\tau^0)} = \log \frac{\prod_{i=1}^{N} p_\theta(\tau^{i-1}|\tau^i)}{\prod_{i=1}^{N} p_{\theta^{\text{old}}}(\tau^{i-1}|\tau^i)} \frac{q(\tau^N)}{q(\tau^N)}$$

The last equality comes from the fact that the reverse and forward processes coincide when the diffusion process reaches equilibrium, with $q(\tau^N)$ being Gaussian that will be canceled out. In addition, taking the expectation of this log density ratio over $\tau^{1:N} \sim q(\cdot|\tau^0)$ does not change anything since it is still ratio of marginal densities. Thus, we can rewrite Eq. (6) as

$$\max_{\theta} \mathbb{E}_{\tau^0 \sim p_E} \left[ \log \sigma \left( \mathbb{E}_{\tau^{1:N} \sim q(\tau^{1:N}|\tau^0)} \log \frac{\prod_{i=1}^{N} p_\theta(\tau^{i-1}|\tau^i)}{\prod_{i=1}^{N} p_{\theta^{\text{old}}}(\tau^{i-1}|\tau^i)} \right) \right] = \max_{\theta} \mathbb{E}_{\tau^0 \sim p_E} \left[ f_\theta(\tau^0) \right]. \tag{7}$$

Finally, applying Jensen's inequality to the concave function $\log \sigma(\cdot)$, we can derive a lower bound of $f$:

$$f_\theta(\tau^0) \geq \mathbb{E}_{i,\tau^i} \left[ \log \sigma \left( N \cdot \left[ \text{KL}\big( q(\tau^{i-1}|\tau^i, \tau^0) \big\| p_{\theta^{\text{old}}}(\tau^{i-1}|\tau^i) \big) - \text{KL}\big( q(\tau^{i-1}|\tau^i, \tau^0) \big\| p_\theta(\tau^{i-1}|\tau^i) \big) \right] \right) \right] \tag{8}$$

where $i \sim \mathcal{U}(1, N), \tau^i \sim q(\tau^i|\tau^0)$.

Since $q(\tau^{i-1}|\tau^i, \tau^0)$ is Gaussian by Bayes' rule, and both $p_\theta(\tau^{i-1}|\tau^i)$ and $p_{\theta^{\text{old}}}(\tau^{i-1}|\tau^i)$ are parameterized as Gaussians, all KL divergences in Eq. (8) admit closed-form expressions. Following [15], we do not predict the mean $\mu_\theta$ of $p_\theta$ directly. Instead, we train a noise prediction network $\epsilon_\theta$, under which the mean is given by: $\mu_\theta(\tau^i, i) = \frac{1}{\sqrt{1-\beta_i}} \left( \tau^i - \frac{\beta_i}{\sqrt{1-\alpha_i}} \epsilon_\theta(\tau^i, i) \right)$. Substituting this into Eq. (8) yields the first term of the $\mathcal{L}_{\text{DPAIL}}^{(1)}$ objective as:

$$\mathcal{L}_{\text{DPAIL}}^{(1)}(\theta, \theta^{\text{old}}, \tau^0) := \mathbb{E}_{i,\epsilon} \left[ \log \sigma \left( N \cdot \big( \|\epsilon - \epsilon_{\theta^{\text{old}}}(\tau^i, i)\|^2 - \|\epsilon - \epsilon_\theta(\tau^i, i)\|^2 \big) \right) \right]. \tag{9}$$

Here, $\epsilon_{\theta^{\text{old}}}$ is the noise prediction network for $p_{\theta^{\text{old}}}$, $\epsilon \sim \mathcal{N}(0, I)$, $i \sim \mathcal{U}(1, N)$, and $\tau^i = \sqrt{\alpha_i}\tau^0 + (1 - \alpha_i)\epsilon$.

For the second term of Eq. (4) that involves generative sample $\bar{\tau}^0$ from $p_{\theta^{\text{old}}}$, the lower bound of the objective is derived similarly due to the symmetry of the sigmoid function $1 - \sigma(\log \frac{p_\theta}{p_{\theta^{\text{old}}}}) = \sigma(\log \frac{p_{\theta^{\text{old}}}}{p_\theta})$:

$$\mathcal{L}_{\text{DPAIL}}^{(2)}(\theta, \theta^{\text{old}}, \bar{\tau}^0) := \mathbb{E}_{i,\epsilon} \left[ \log \sigma \left( N \cdot \big( \|\epsilon - \epsilon_\theta(\bar{\tau}^i, i)\|^2 - \|\epsilon - \epsilon_{\theta^{\text{old}}}(\bar{\tau}^i, i)\|^2 \big) \right) \right]. \tag{10}$$

Eq. (9) encourages accurate noise prediction for noised expert data by maximizing the gap in prediction errors between $\epsilon_\theta$ and $\epsilon_{\theta^{\text{old}}}$. In contrast, Eq. (10) discourages accurate noise prediction for noised generative data in $\epsilon_\theta$. Detailed derivations are provided in Appendix A.

**Monotonic improvement** We employ the lower bound of Eq. (8),

$$\mathbb{E}_{\tau^0 \sim p_E} \left[ \mathcal{L}_{\text{DPAIL}}^{(1)}(\theta, \theta^{\text{old}}, \tau^0) \right] + \mathbb{E}_{\bar{\tau}^0 \sim p_{\theta^{\text{old}}}} \left[ \mathcal{L}_{\text{DPAIL}}^{(2)}(\theta, \theta^{\text{old}}, \bar{\tau}^0) \right],$$

as a surrogate objective. Importantly, our algorithm becomes an instance of minorization-maximization algorithm since (1) we optimize with a surrogate objective which is a lower bound of the original objective (2) the lower bound is tight at the convergence $\theta^{\text{old}} = \theta$. This guarantees monotonic improvement of the original training objective. Further explanations are provided in Appendix A.

---

**Algorithm 1** DPAIL

---

**Input:** expert trajectories $\mathcal{D}_E = \{\tau_n\}_{n=1}^{N_E}$

Randomly initialize $p_{\theta_0}$ and set $p_{\theta^{\text{old}}} \leftarrow p_{\theta_0}$

**for** $k \in [0, \dots, K]$ **do**

  Collect trajectories $\mathcal{D}_{\theta^{\text{old}}} = \{\bar{\tau}_n\}_{n=1}^{N_{\theta^{\text{old}}}}$ using $p_{\theta^{\text{old}}}$ by interacting with environment

  Update $\theta_{k+1}$ by optimizing the following loss in Eq. (9) and Eq. (10):

  $$\theta_{k+1} = \arg\max_\theta \mathbb{E}_{\tau^0 \sim \mathcal{D}_E}[\mathcal{L}_{\text{DPAIL}}^{(1)}(\theta, \theta^{\text{old}}, \tau^0)] + \mathbb{E}_{\bar{\tau}^0 \sim \mathcal{D}_{\theta^{\text{old}}}}[\mathcal{L}_{\text{DPAIL}}^{(2)}(\theta, \theta^{\text{old}}, \bar{\tau}^0)].$$

  $p_{\theta^{\text{old}}} \leftarrow p_{\theta_{k+1}}$

**end for**

---

**Practical implementation**   In our approach, the diffusion policy generates fixed-horizon sub-trajectories of length $H$, instead of producing full trajectories. We then execute the resulting $H$ actions sequentially in the environment. To condition on the current state at the start of the sampling process, we overwrite the corresponding state variable at every diffusion step with the current observed state. The detailed procedures for action execution and sampling appear in Algorithm 2 and 3 in Appendix. Since there is a discrepancy between the trajectories sampled entirely from the generator $p_\pi$ and those obtained from the environment, we clip the denoising error $\|\epsilon - \epsilon_{\theta^{\text{old}}}(\bar{\tau}^i, i)\|^2$ to prevent it from being too large. The complete DPAIL algorithm is summarized in Algorithm 1. Additional implementation details can be found in Appendix D.

**Comparison with DRAIL**   DRAIL [18] was introduced to incorporate diffusion models into the discriminator in GAIL. It employs conditional diffusion models for the discriminator, with the loss function given by:

$$\mathcal{L}_{\text{diff}}(\tau^0, c) := \mathbb{E}_{i,\epsilon}[\|\epsilon_\phi(\tau^i, i|c) - \epsilon\|^2] \tag{11}$$

where $c \in \{c^+, c^-\}$ denotes real or fake labels. This loss function forms the basis of the diffusion-based discriminator $D_\phi : \mathcal{T} \to [0, 1]$,

$$D_\phi(\tau^0) := \sigma\big(\mathcal{L}_{\text{diff}}(\tau^0, c^-) - \mathcal{L}_{\text{diff}}(\tau^0, c^+)\big),$$

which is trained using a binary cross-entropy loss to predict 1 for expert samples and 0 for generated samples. The overall diffusion training objective of DRAIL is:

$$\mathbb{E}_{\tau^0 \sim p_E}[\log \sigma\big(\mathcal{L}_{\text{diff}}(\tau^0, c^-) - \mathcal{L}_{\text{diff}}(\tau^0, c^+)\big)] + \mathbb{E}_{\bar{\tau}^0 \sim p_\pi}[\log \sigma\big(\mathcal{L}_{\text{diff}}(\bar{\tau}^0, c^+) - \mathcal{L}_{\text{diff}}(\bar{\tau}^0, c^-)\big)]. \tag{12}$$

Both DRAIL (Eq. (12)) and DPAIL (Eq. (9)) rely on differences in noise predictions errors within a sigmoid function. In DRAIL, this difference arise from comparing predictions conditioned on two different class labels $c^+$ and $c^-$, using the same noise prediction network $\epsilon_\phi$. In contrast, DPAIL computes the difference using two distinct noise prediction networks, $\epsilon_\theta$ and $\epsilon_{\theta^{\text{old}}}$. Despite this similarity in formulation, the two approaches differ fundamentally in how the diffusion model is used. DRAIL trains a unimodal Gaussian policy via RL, using the diffusion models purely as a reward signal, which limits its ability to represent multi-modal distributions. DPAIL, on the other hand, directly leverages the generative capacity of diffusion models to reproduce the diverse behavior in the expert demonstrations.

**Extension to latent-conditioned diffusion policies**   Unlike InfoGAIL [20], our method does not inherently learn latent representations, which makes it difficult to guide the policy's behavior toward specific modes. However, the DPAIL framework can be extended to incorporate latent variables $z$ for mode conditioning, as this is compatible with our derived lower bound. To enable this, we define a latent-conditioned diffusion model where the joint probability of the generative process $p_\theta(\tau^{0:N}, z)$ factorizes as:

$$p_\theta(\tau^{0:N}, z) = q(\tau^N)q(z)\prod_{i=1}^{N} p_\theta(\tau^{i-1}|\tau^i, z)$$

where $q(z)$ is the latent prior and $q(\tau^N)$ is the noise prior. To derive the corresponding tractable lower bound, we follow a similar procedure to the main DPAIL formulation; the detailed derivation is provided in Appendix F.

# 5 Experiments

In this section, we evaluate our method across navigation and control tasks, including Maze2d and MuJoCo environments. We begin with quantitative results that demonstrate its effectiveness at modeling multi-modal expert demonstrations. Next, we provide qualitative trajectory visualization to illustrate its ability to reproduce diverse behaviors. We then investigate how performance varies with the size of the expert dataset and the number of behavior modes. Finally, we wrap up with the analysis on the effects of the trajectory horizon $H$ and the number of diffusion sampling steps $N$.

## 5.1 Experimental Setup

**Environment**   We conduct experiments in six environments: **(1) HalfCheetah-v3** and **(2) Walker2d-v3**: The goal of these environments is to control a robot to move in the desired direction, including running forward and backward. **(3) Ant-v3**: In this setup, a quadruped ant robot is tasked with moving in one of four directions, including forward, backward, left, or right. **(4) AntGoal-v3**: This environment requires a quadruped ant robot to navigate to a target position. We define eight target locations, evenly distributed along a circle with a radius of 20. The target position is not observable in the state representation. **(5) maze2d-medium-v1**: A point robot is tasked with navigating to one of three goal positions in a medium-sized maze. **(6) maze2d-large-v1**: Similar to the medium-sized maze task, but in a larger maze with five goal positions.

**Multi-modal demonstration dataset**   For MuJoCo environments, we pre-train $M$ expert policies using SAC, where each policy corresponds to one of $M$ behavior modes. We then sample $K$ sets of expert demonstrations using these pre-trained policies, with each set consisting of 10 trajectories in MuJoCo. For Maze2d environments, we utilize the D4RL [11] dataset to collect demonstrations consisting of trajectories from initial positions to goal positions. Specifically, we use 15 episodes for maze2d-medium-v1 and 30 episodes for maze2d-large-v1.

**Baselines**   We compare our method against the following baselines:

- **BC** [4, 24] learns a Gaussian policy via supervised learning, a mapping from observed states to the corresponding expert actions.
- **Diffusion** [17, 7] trains diffusion policy models to predict action sequence conditioned on the state via supervised learning.
- **GAIL** [14] learns a Gaussian policy by jointly training a generator and a discriminator. The discriminator tries to distinguish trajectories produced by the policy from expert demonstrations, while the policy is optimized to fool the discriminator.
- **DiffAIL** [34] integrates diffusion models into AIL by using the diffusion model loss as a reward. It employs an unconditional diffusion model in the state-action reconstruction loss. Unlike GAIL, the reward does not come from the estimated value of (5).
- **DRAIL** [18] combines diffusion models with GAIL by using conditional diffusion models as a discriminator that performs binary classification.
- **InfoGAIL** [20, 13] is an extension of GAIL that trains Gaussian Mixture Models (GMM) to capture multi-modal behaviors by incorporating an unsupervised representation. It uses a uniform categorical distribution as the prior for the GMM.
- **ASAF** [6] is an alternative approach to train a Gaussian policy without policy optimization in AIL.

## 5.2 Experimental Results

To compare the performances of methods, we use normalized scores for MuJoCo tasks and success rates for Maze2d tasks. For MuJoCo tasks, we compute returns across all modes, select the maximum value, and normalize it relative to the corresponding expert performance, following [10, 7]. This score is high even if the agent captures only a single mode of the expert behavior. In Maze2d tasks, success is defined by whether the agent reaches one of the goals. We also evaluate behavioral diversity by measuring the entropy of the mode index that yields the maximum return during evaluation. In Maze2d, the goal index closest to the agent's final state is used for entropy calculation. Table 1 summarizes the results.

Table 1: Normalized score (*Score*) and entropy (*Entropy*) for MuJoCo and Maze2d tasks. Each experiments is conducted using 5 different random seeds, and we collect 50 episodes for each seed. We report the scores as mean ± standard error.

| Environment | Metrics (↑) | BC | Diffusion | GAIL | DiffAIL | DRAIL | InfoGAIL | ASAF | DPAIL |
|---|---|---|---|---|---|---|---|---|---|
| HalfCheetah-v3 | Score | 0.61 ± 0.18 | **1.01** ± 0.00 | 0.96 ± 0.03 | **0.99** ± 0.01 | 0.89 ± 0.04 | 0.65 ± 0.05 | 0.62 ± 0.08 | **1.02** ± 0.01 |
| | Entropy | 0.23 ± 0.12 | 0.58 ± 0.05 | 0.00 ± 0.00 | 0.00 ± 0.00 | 0.02 ± 0.01 | 0.33 ± 0.14 | 0.33 ± 0.11 | 0.61 ± 0.06 |
| Walker2d-v3 | Score | 0.03 ± 0.00 | 0.58 ± 0.03 | 0.72 ± 0.01 | 0.53 ± 0.10 | 0.33 ± 0.04 | 0.55 ± 0.12 | 0.03 ± 0.00 | **0.78** ± 0.04 |
| | Entropy | 0.26 ± 0.12 | 0.41 ± 0.10 | 0.00 ± 0.00 | 0.06 ± 0.06 | 0.21 ± 0.11 | 0.44 ± 0.11 | 0.40 ± 0.02 | 0.41 ± 0.13 |
| Ant-v3 | Score | 0.18 ± 0.09 | 0.48 ± 0.04 | 0.01 ± 0.00 | 0.02 ± 0.00 | 0.02 ± 0.00 | 0.03 ± 0.02 | 0.01 ± 0.00 | **0.56** ± 0.02 |
| | Entropy | 0.28 ± 0.11 | 1.22 ± 0.05 | 1.19 ± 0.04 | 1.22 ± 0.04 | 1.26 ± 0.04 | 1.04 ± 0.03 | 1.07 ± 0.08 | 1.21 ± 0.02 |
| AntGoal-v3 | Score | 0.04 ± 0.01 | 0.22 ± 0.03 | 0.57 ± 0.02 | 0.35 ± 0.07 | 0.41 ± 0.06 | 0.48 ± 0.03 | 0.02 ± 0.00 | **0.67** ± 0.01 |
| | Entropy | 1.46 ± 0.26 | 1.52 ± 0.13 | 1.51 ± 0.07 | 1.75 ± 0.04 | 1.72 ± 0.03 | 1.78 ± 0.05 | 1.52 ± 0.12 | 1.73 ± 0.10 |
| maze2d-medium-v1 | Score | 0.58 ± 0.17 | 0.93 ± 0.03 | 0.76 ± 0.19 | 0.92 ± 0.05 | 0.98 ± 0.01 | 0.76 ± 0.10 | 0.64 ± 0.09 | **0.99** ± 0.00 |
| | Entropy | 0.66 ± 0.18 | 0.93 ± 0.08 | 0.12 ± 0.12 | 0.00 ± 0.00 | 0.00 ± 0.00 | 0.81 ± 0.05 | 0.75 ± 0.10 | 0.91 ± 0.02 |
| maze2d-large-v1 | Score | 0.61 ± 0.18 | 0.94 ± 0.04 | 0.94 ± 0.01 | 0.96 ± 0.02 | 0.95 ± 0.03 | 0.74 ± 0.08 | 0.79 ± 0.10 | **1.00** ± 0.00 |
| | Entropy | 0.61 ± 0.26 | 0.95 ± 0.20 | 0.10 ± 0.11 | 0.00 ± 0.00 | 0.45 ± 0.13 | 1.05 ± 0.06 | 0.54 ± 0.14 | 1.04 ± 0.11 |
| Average | Score | 0.34 ± 0.11 | 0.69 ± 0.11 | 0.66 ± 0.13 | 0.63 ± 0.14 | 0.59 ± 0.14 | 0.53 ± 0.10 | 0.35 ± 0.13 | **0.83** ± 0.06 |
| | Entropy | 0.58 ± 0.19 | 0.93 ± 0.15 | 0.63 ± 0.25 | 0.50 ± 0.28 | 0.61 ± 0.26 | 0.98 ± 0.20 | 0.76 ± 0.16 | 0.90 ± 0.19 |

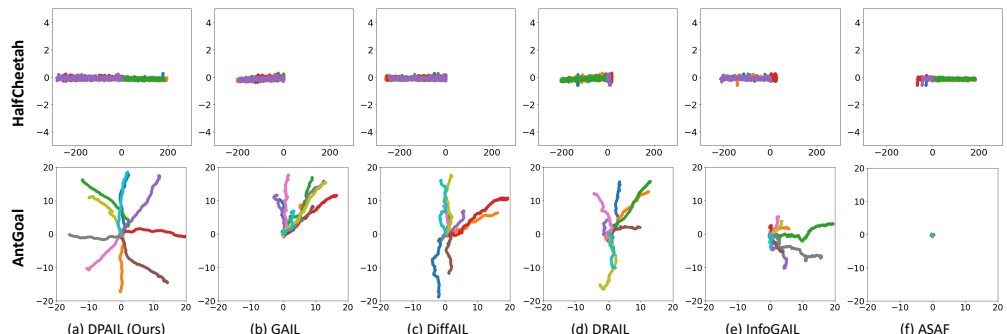

Figure 1: Learned behaviors of baseline methods and DPAIL (ours) in MuJoCo tasks. The first row shows HalfCheetah-v3, where the task is to move forward and backward (±x-axis) from a randomly initialized position around (0, 0). Each plot illustrates five different trajectories generated from the same policy. The second row shows AntGoal-v3, where the task is to reach one of eight target positions distributed around a circle from a randomly initialized position near (0, 0). Each plot illustrates ten different trajectories generated by the same policy.

In HalfCheetah-v3, all methods demonstrate strong performance except for BC. Specifically, GAIL, DiffAIL, and DRAIL reach expert-level performance, but, yield zero or near-zero entropy, suggesting they collapse to a single mode. While InfoGAIL attains higher entropy, it shows relatively lower performance, which indicate the challenges of unsupervised representation learning. In contrast, Diffusion and DPAIL achieve high performance and high entropy.

For more complex tasks with numerous behavior modes, such as Ant-v3 and AntGoal-v3, most baseline methods (including Diffusion) perform poorly. GAIL, DiffAIL, DRAIL, and ASAF fail to train, resulting in low returns and high entropy. These observations suggest instability when learning from multiple modes—agents frequently oscillate among modes and fail to learn a stable policy. Notably, training failures frequently yield nearly random behaviors, which lead to high entropy. For example, most baseline methods fail on Ant-v3, culminating in low performance but high entropy. Although InfoGAIL leverages the GMM to model diverse behaviors, it also exhibits low performance. This suggests that combining RL policy training with unsupervised representation learning remains as a challenging problem.

Meanwhile, Diffusion, an naive offline approach, sometimes outperforms several online methods in terms of returns and entropy (e.g., in Ant-v3 and Maze2d), likely due to the stability of its training process to effectively handle multiple modes. However, it struggles to perfectly imitate expert behaviors with limited datasets. In contrast, DPAIL consistently achieves strong performance alongside high entropy across all tasks. Even in environments with many behavior modes, DPAIL successfully avoids mode-collapse due to the capability of diffusion models, and successfully learn expert behaviors through the adversarial training. This demonstrates the effectiveness of our approach.

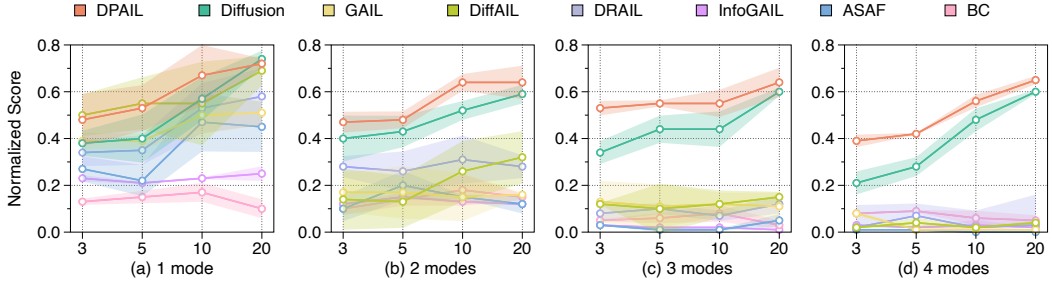

Figure 2: Performance with respect to the number of demonstrations and modes. The x-axis denotes the number of expert trajectories. The graph depicts the average score across 5 seeds with standard error in Ant-v3 environment.

To qualitatively analyze the learned behaviors of our algorithm and baselines, we visualize trajectories sampled from policies in Figure 1. In MuJoCo tasks, DPAIL demonstrates diverse and successful behaviors across tasks. In contrast, most algorithms exhibit mode-collapse, where their learned behaviors concentrated in only a few modes. Although InfoGAIL has high entropy, it struggles to imitate even a single mode. We present additional qualitative results of the learned behaviors in the Maze tasks in Appendix E.

## 5.3 Further Analysis

**Impact of the number of demonstrations and modes** To further assess DPAIL, we analyze its performance across different demonstration dataset sizes ($\{3, 5, 10, 20\}$ trajectories) and the number of modes, ($\{1, 2, 3, 4\}$) in Ant-v3 environment, where most baseline methods fail to learn policies. The results are presented in Figure 2.

In single-mode settings, all methods achieve stronger performance than in four-mode settings. However, as the number of modes increases, most algorithms (excluding Diffusion and DPAIL) experience significant performance degradation, ultimately failing to train. These findings highlight the limitations of traditional AIL methods in handling multi-modal expert demonstrations.

While both Diffusion and DPAIL exhibit performance declines as the number of modes increases, they do not fail to train policies. They maintain minimum normalized scores of 0.2 (Diffusion) and 0.4 (DPAIL) in the 3 trajectories demonstration and 4-modes setting. Notably, DPAIL consistently outperforms Diffusion when the number of demonstrations is limited. However, once the number of demonstrations becomes sufficiently large (e.g. 20 trajectories), Diffusion performs comparably to DPAIL. These results demonstrates the effectiveness of DPAIL for scenarios requiring imitation from limited, multi-modal demonstrations. Extended quantitative analyses supporting Figure 2 are provided Table 5 in Appendix.

**Inference compute versus planning horizon $H$** DPAIL incurs higher inference cost due to $N$-step diffusion sampling for action generation. To reduce this cost, we generate $H$-step sub-trajectories and execute the resulting $H$ actions sequentially, requiring diffusion sampling only every $H$ time-steps.

However, longer $H$ introduces the compounding errors between predicted and the true state transitions. Figure 3 presents the trade-off between inference time per action (Time Cost) and performance (Score) across different values of $H$. As shown, shorter horizon yield better performance at higher time cost, while longer horizons reduce inference overhead bur degrade performance due to the accumulated prediction errors.

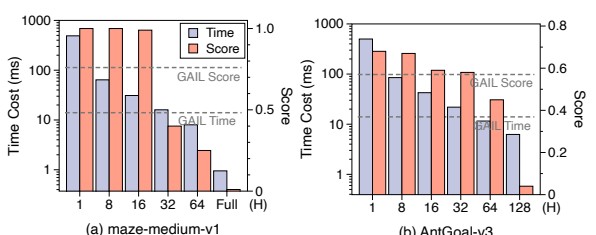

Figure 3: Inference compute versus planning horizon $H$

**Diversity versus diffusion sampling step** $N$ The number of diffusion steps $N$ directly influences the model's ability to approximate the expert's multi-modal distribution. To evaluate this effect, we vary $N$ and report both entropy and task performance in Figure 4. As $N$ increases, entropy rises, indicating richer behavioral diversity, while performance remains nearly unchanged. This findings indicate that at low $N$, the policy captures only a few dominated modes, whereas higher $N$ values allow it to represent a broader spectrum of behaviors.

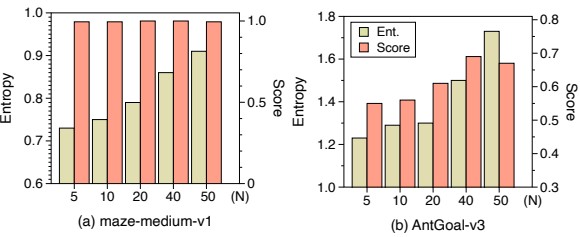

Figure 4: Diversity versus diffusion sampling step $N$

# 6   Conclusion and Discussion

We presented Diffusion Policy for Adversarial Imitation Learning (DPAIL), a framework integrates diffusion models into Adversarial Imitation Learning (AIL) to effectively capture multi-modal expert behaviors. DPAIL avoids the mode-collapse and instability issues faced by many existing methods when learning from multi-modal demonstrations. Extensive experiments on MuJoCo and Maze2d tasks demonstrate that DPAIL consistently achieves high returns and maintains high diversity, even in challenging environments with numerous expert modes.

# 7   Limitations and Future Work

While our approach effectively models multi-modal behaviors using diffusion models, it requires more computation costs for action decision compared to other baselines. This overhead arises from the iterative sampling process used by diffusion models. Several works [26, 12] have proposed methods to accelerate sampling in diffusion process. Incorporating these techniques into DPAIL is a promising direction to enhance its efficiency. Furthermore, while we demonstrate that DPAIL is compatible with latent-conditioned diffusion policies, we did not provide an empirical validation. We leave this as future work. Additionally, other directions exist for guiding policy behavior toward specific modes beyond training a latent-conditioned policy. For instance, recent research [29] has explored guidance mechanisms that steer the reverse diffusion process toward desired outcomes. Extending our framework with such mechanisms could further improve controllability in multi-modal settings.

## Broader Impacts

The potential applications of our method extend across various fields, particularly in robotics and industrial automation, where learning from demonstrations is crucial. In these domains, our approach can significantly enhance autonomous decision-making by enabling effective imitation learning from multi-modal expert demonstrations, where conventional algorithms often fail. By successfully capturing this behavioral diversity, our work contributes to the advancement of more capable and reliable AI for real-world deployment.

## Acknowledgments

This work was supported by Institute for Information & Communications Technology Promotion (IITP), funded by MSIT, through the Information Technology Research Center (ITRC) program and other projects (No.RS-2024-00397310, Development of an AI Simulator for Creating Transparent Compounds that Can Be Altered for Tactile Sensation; No.RS-2024-00457882, AI Research Hub Project; No.RS-2022-II220311, Development of Goal-Oriented Reinforcement Learning Techniques for Contact-Rich Robotic Manipulation of Everyday Objects; No.RS-2020-II200940, Foundations of Safe Reinforcement Learning and Its Applications to Natural Language Processing; No.RS-2019-II190075, Artificial Intelligence Graduate School Program (KAIST); IITP-2025-RS-2024-00436857).

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

# A  Details of the Derivation

## A.1  Derivations of Eq. (9) and Eq. (10)

In this section, we provide detailed derivations of Eq. (9) and Eq. (10). Given the expert sample $\tau^0 \sim p_E$, the reverse process $p_\theta(\tau^{i-1}|\tau^i)$, and the corresponding forward process $q(\tau^{1:N}|\tau^0)$,

$$\mathbb{E}_{\tau^0 \sim p_E}\left[\log \sigma\left(\mathbb{E}_{\tau^{1:N} \sim q(\tau^{1:N}|\tau^0)} \log \frac{\prod_{i=1}^N p_\theta(\tau^{i-1}|\tau^i)}{\prod_{i=1}^N p_{\theta^{\text{old}}}(\tau^{i-1}|\tau^i)}\right)\right]$$

$$= \mathbb{E}_{\tau^0 \sim p_E}\left[\log \sigma\left(\mathbb{E}_{\tau^{1:N} \sim q(\tau^{1:N}|\tau^0)} \sum_{i=1}^N \log \frac{p_\theta(\tau^{i-1}|\tau^i)}{p_{\theta^{\text{old}}}(\tau^{i-1}|\tau^i)}\right)\right]$$

$$= \mathbb{E}_{\tau^0 \sim p_E}\left[\log \sigma\left(\sum_{i=1}^N \mathbb{E}_{\tau^{1:N} \sim q(\tau^{1:N}|\tau^0)} \log \frac{p_\theta(\tau^{i-1}|\tau^i)}{p_{\theta^{\text{old}}}(\tau^{i-1}|\tau^i)}\right)\right]$$

$$= \mathbb{E}_{\tau^0 \sim p_E}\left[\log \sigma\left(\sum_{i=1}^N \mathbb{E}_{\tau^{i-1},\tau^i \sim q(\tau^i|\tau^0)q(\tau^{i-1}|\tau^i,\tau^0)} \log \frac{p_\theta(\tau^{i-1}|\tau^i)}{p_{\theta^{\text{old}}}(\tau^{i-1}|\tau^i)}\right)\right]$$

$$\geq \mathbb{E}_{\tau^0 \sim p_E}\left[\mathbb{E}_{i \sim \mathcal{U}(1,N),\tau^i \sim q(\tau_i|\tau_0)} \log \sigma\left(N \mathbb{E}_{\tau^{i-1} \sim q(\tau^{i-1}|\tau^i,\tau^0)} \log \frac{p_\theta(\tau^{i-1}|\tau^i)}{p_{\theta^{\text{old}}}(\tau^{i-1}|\tau^i)}\right)\right] \quad \text{(concavity)}$$

$$= \mathbb{E}_{\tau^0 \sim p_E}\left[\mathbb{E}_{i \sim \mathcal{U}(1,N),\tau^i \sim q(\tau_i|\tau_0)} \log \sigma\left(N \mathbb{E}_{\tau^{i-1} \sim q(\tau^{i-1}|\tau^i,\tau^0)} \log \frac{q(\tau^{i-1}|\tau^i,\tau^0)}{p_{\theta^{\text{old}}}(\tau^{i-1}|\tau^i)} - \log \frac{q(\tau^{i-1}|\tau^i,\tau^0)}{p_\theta(\tau^{i-1}|\tau^i)}\right)\right]$$

$$\tag{13}$$

The posterior $q(\tau^{i-1}|\tau^i, \tau^0)$ is tractable via Bayes' rule, since both $q(\tau^{i-1}|\tau^0)$ and $q(\tau^i|\tau^{i-1}, \tau^0) = q(\tau^i|\tau^{i-1})$ are Gaussian distributions. The posterior $q(\tau^{i-1}|\tau^i, \tau^0)$ can be written as the following:

$$q(\tau^{i-1}|\tau^i, \tau_0) = \mathcal{N}(\tilde{\mu}_i(\tau^i, \tau^0), \tilde{\beta}_i I)$$

where $\tilde{\mu}_i(\tau^i, \tau^0) := \frac{1}{\sqrt{1-\beta_i}}\left(\tau^i - \frac{\beta_i}{\sqrt{1-\alpha_i}}z(\tau^i, \tau^0)\right)$, $z(\tau^i, \tau^0) := \frac{\tau^i - \sqrt{\alpha_i}\tau^0}{\sqrt{1-\alpha_i}} = \epsilon$. If we parameterize the model $p_\theta$ as $\mathcal{N}(\mu_\theta(\tau^i, i), \sigma_i^2 I)$, then the KL divergence between these isotropic Gaussians reduces to the squared error between the means: $||\tilde{\mu}_i(\tau^i, \tau^0) - \mu_\theta(\tau_i, i)||^2$. Ho et al. [15] proposes to parameterize $\mu_\theta$ using a predicted noise $\epsilon_\theta$, such as:

$$\mu_\theta(\tau^i, i) = \frac{1}{\sqrt{1-\beta_i}}\left(\tau^i - \frac{\beta_i}{\sqrt{1-\alpha_i}}\epsilon_\theta(\tau^i, i)\right)$$

Substituting this into the squared error gives:

$$\left\|\tilde{\mu}_i(\tau^i, \tau^0) - \mu_\theta(\tau^i, i)\right\|^2 = \left\|\frac{1}{\sqrt{1-\beta_i}}\left(\tau^i - \frac{\beta_i}{\sqrt{1-\alpha_i}}z(\tau^i, \tau^0)\right) - \mu_\theta(\tau^i, i)\right\|^2$$

$$= \left\|\frac{1}{\sqrt{1-\beta_i}}\left(\tau^i - \frac{\beta_i}{\sqrt{1-\alpha_i}}\epsilon\right) - \frac{1}{\sqrt{1-\beta_i}}\left(\tau^i - \frac{\beta_i}{\sqrt{1-\alpha_i}}\epsilon_\theta(\tau^i, i)\right)\right\|^2$$

$$= \frac{\beta_i}{(1-\beta_i)(1-\alpha_i)}\left\|\epsilon - \epsilon_\theta(\tau^i, i)\right\|^2$$

Therefore, minimizing the KL divergence is equivalent to minimizing $||\epsilon - \epsilon_\theta(\tau^i, i)||^2$. Substituting this into Eq. 13 leads to the following equation:

$$\therefore \mathbb{E}_{\tau^0, i, \epsilon}\left[\log \sigma\left(N \cdot C_i \left(||\epsilon - \epsilon_{\theta^{\text{old}}}(\tau^i, i)||^2 - ||\epsilon - \epsilon_\theta(\tau^i, i)||^2\right)\right)\right].$$

Here, $C_i = \frac{\beta_i}{(1-\beta_i)(1-\alpha_i)}$, $\epsilon \sim \mathcal{N}(0, I)$, and $\tau^i \sim q(\tau^i|\tau^0)$, thus $\tau^i = \sqrt{\alpha_i}\tau^0 + (1-\alpha_i)\epsilon$. Following Wallace et al. [33], we consider the weight $N \cdot C_i$ as a fixed constant value over $i$ in practical implementation. Similarly, given the generative sample $\bar{\tau}^0 \sim p_{\theta^{\text{old}}}$, the reverse process $p_{\theta^{\text{old}}}(\bar{\tau}^{i-1}|\bar{\tau}^i)$, and the corresponding forward process $q_{\theta^{\text{old}}}(\bar{\tau}^{1:N}|\bar{\tau}^0) = q(\bar{\tau}^{1:N}|\bar{\tau}^0)$ (since they have the same

variance schedule),

$$\mathbb{E}_{\bar{\tau}^0 \sim p_{\theta^{\mathrm{old}}}} \left[ \log \sigma \left( \mathbb{E}_{\bar{\tau}^{1:N} \sim q(\tau^{1:N}|\bar{\tau}^0)} \log \frac{\prod_{i=1}^{N} p_{\theta^{\mathrm{old}}}(\bar{\tau}^{i-1}|\bar{\tau}^i)}{\prod_{i=1}^{N} p_\theta(\bar{\tau}^{i-1}|\bar{\tau}^i)} \right) \right]$$

$$= \mathbb{E}_{\bar{\tau}^0 \sim p_{\theta^{\mathrm{old}}}} \left[ \log \sigma \left( \mathbb{E}_{\bar{\tau}^{1:N} \sim q(\bar{\tau}^{1:N}|\bar{\tau}^0)} \sum_{i=1}^{N} \log \frac{p_{\theta^{\mathrm{old}}}(\bar{\tau}^{i-1}|\bar{\tau}^i)}{p_\theta(\bar{\tau}^{i-1}|\bar{\tau}^i)} \right) \right]$$

$$= \mathbb{E}_{\bar{\tau}^0 \sim p_{\theta^{\mathrm{old}}}} \left[ \log \sigma \left( \sum_{i=1}^{N} \mathbb{E}_{\bar{\tau}^{1:N} \sim q(\bar{\tau}^{1:N}|\bar{\tau}^0)} \log \frac{p_{\theta^{\mathrm{old}}}(\bar{\tau}^{i-1}|\bar{\tau}^i)}{p_\theta(\bar{\tau}^{i-1}|\bar{\tau}^i)} \right) \right]$$

$$= \mathbb{E}_{\bar{\tau}^0 \sim p_{\theta^{\mathrm{old}}}} \left[ \log \sigma \left( \sum_{i=1}^{N} \mathbb{E}_{\bar{\tau}^{i-1}, \bar{\tau}^i \sim q(\bar{\tau}^i|\bar{\tau}^0) q(\bar{\tau}^{i-1}|\bar{\tau}^i, \bar{\tau}^0)} \log \frac{p_{\theta^{\mathrm{old}}}(\bar{\tau}^{i-1}|\bar{\tau}^i)}{p_\theta(\bar{\tau}^{i-1}|\bar{\tau}^i)} \right) \right]$$

$$\geq \mathbb{E}_{\bar{\tau}^0 \sim p_{\theta^{\mathrm{old}}}} \left[ \mathbb{E}_{i \sim \mathcal{U}(1,N), \bar{\tau}^i \sim q(\bar{\tau}_i|\bar{\tau}_0)} \log \sigma \left( N \mathbb{E}_{\bar{\tau}^{i-1} \sim q(\bar{\tau}^{i-1}|\bar{\tau}^i, \bar{\tau}^0)} \log \frac{p_{\theta^{\mathrm{old}}}(\bar{\tau}^{i-1}|\bar{\tau}^i)}{p_\theta(\bar{\tau}^{i-1}|\bar{\tau}^i)} \right) \right] \qquad \text{(concavity)}$$

$$= \mathbb{E}_{\bar{\tau}^0 \sim p_{\theta^{\mathrm{old}}}} \left[ \mathbb{E}_{i \sim \mathcal{U}(1,N), \tau^i \sim q(\bar{\tau}_i|\bar{\tau}_0)} \log \sigma \left( N \mathbb{E}_{\bar{\tau}^{i-1} \sim q(\bar{\tau}^{i-1}|\bar{\tau}^i, \bar{\tau}^0)} \log \frac{q(\bar{\tau}^{i-1}|\bar{\tau}^i, \bar{\tau}^0)}{p_\theta(\bar{\tau}^{i-1}|\bar{\tau}^i)} - \log \frac{q(\bar{\tau}^{i-1}|\bar{\tau}^i, \bar{\tau}^0)}{p_{\theta^{\mathrm{old}}}(\bar{\tau}^{i-1}|\bar{\tau}^i)} \right) \right]$$

$$\therefore \mathbb{E}_{\bar{\tau}^0, i, \epsilon} \left[ \log \sigma \left( N \cdot C_i \left( \|\epsilon - \epsilon_\theta(\bar{\tau}^i, i)\|^2 - \|\epsilon - \epsilon_{\theta^{\mathrm{old}}}(\bar{\tau}^i, i)\|^2 \right) \right) \right].$$

$C_i = \frac{\beta_i}{(1-\beta_i)(1-\alpha_i)}$, $\epsilon \sim \mathcal{N}(0, I)$ and $\bar{\tau}^i \sim q(\bar{\tau}^i|\bar{\tau}^0)$, thus, $\bar{\tau}^i = \sqrt{\alpha_i}\bar{\tau}^0 + (1 - \alpha_i)\epsilon$.

## A.2 Monotonic Improvement

In this section, we show that maximizing the surrogate objective guarantees a monotonic improvement in the original training objective. Revisiting Eq. (8), we denote the lower bound by $g_{\theta_k, \theta}(\tau^0)$, where $\theta_k = \theta^{\mathrm{old}}$:

$$f_\theta(\tau^0) = \log \sigma \left( \mathbb{E}_{\tau^{1:N} \sim q(\tau^{1:N}|\tau^0)} \log \frac{\prod_{i=1}^{N} p_\theta(\tau^{i-1}|\tau^i)}{\prod_{i=1}^{N} p_{\theta_k}(\tau^{i-1}|\tau^i)} \right)$$

$$\geq \mathbb{E}_{i \sim \mathcal{U}(1,N), \tau^i \sim q(\tau_i|\tau_0)} \log \sigma \left( N \mathbb{E}_{\tau^{i-1} \sim q(\tau^{i-1}|\tau^i, \tau^0)} \log \frac{p_\theta(\tau^{i-1}|\tau^i)}{p_{\theta_k}(\tau^{i-1}|\tau^i)} \right) = g_{\theta_k, \theta}(\tau^0).$$

Since $\log \sigma(\cdot)$ is a concave function, the lower bound $g_{\theta_k, \theta}(\tau^0)$ is always less than or equal to the original $f_\theta(\tau^0)$ for all $\theta \in \Theta$: (1) $f_\theta(\tau^0) \geq g_{\theta_k, \theta}(\tau^0)$. Moreover, when $\theta = \theta_k$, the log term in both sides becomes zero, yielding (2) $f_{\theta_k}(\tau^0) = g_{\theta_k, \theta_k}(\tau^0) = \log 1/2$.

If we maximize $g_{\theta_k, \theta}(\tau^0)$ instead of $f_\theta(\tau^0)$:

$$\theta_{k+1} = \arg\max_{\theta \in \Theta} g_{\theta_k, \theta}(\tau^0),$$

then the following inequality holds:

$$f_{\theta_{k+1}}(\tau^0) \geq g_{\theta_k, \theta_{k+1}}(\tau^0) \geq g_{\theta_k, \theta_k}(\tau^0) = f_{\theta_k}(\tau^0).$$

Therefore, this procedure guarantees monotonic improvement of the original objective.

## A.3 Comparison with DPO-Diffusion

DPO-Diffusion [33] is an algorithm for aligning diffusion models with human preferences by considering ranked pairs $(\tau_w^0, \tau_l^0)$ that indicate a preference for $\tau_w^0$ over $\tau_l^0$. Built upon the Bradley-Terry (BT) model and the bijectivity between reward and policy, DPO-Diffusion optimizes the diffusion model $p_\theta$ with a reference distribution $p_{\mathrm{ref}}$ via:

$$\mathbb{E}_{(\tau_w^0, \tau_l^0)} \log \sigma \left( \mathbb{E}_{(\tau_w^{1:N}, \tau_l^{1:N})} \left[ \log \frac{p_\theta(\tau_w^{1:N})}{p_{\mathrm{ref}}(\tau_w^{1:N})} - \log \frac{p_\theta(\tau_l^{1:N})}{p_{\mathrm{ref}}(\tau_w^{1:N})} \right] \right), \tag{14}$$

where $\tau_w^{1:N} \sim q(\cdot|\tau_w^0)$ and $\tau_l^{1:N} \sim q(\cdot|\tau_l^0)$. This objective function can be reformulated in terms of noise prediction as:

$$\mathbb{E}_{(\tau_w^0, \tau_l^0, i)} \log \sigma \Big( N(\|\epsilon_w - \epsilon_{\text{ref}}(\tau_w^i, i)\|^2 - \|\epsilon_w - \epsilon_\theta(\tau_w^i, i)\|^2 + \|\epsilon_l - \epsilon_\theta(\tau_l^i, i)\|^2 - \|\epsilon_l - \epsilon_{\text{ref}}(\tau_l^i, i)\|^2) \Big), \quad (15)$$

where $\epsilon_w$ and $\epsilon_l$ correspond to $\tau_w^i$ and $\tau_l^i$, $\epsilon_\theta$ and $\epsilon_{\text{ref}}$ are the noise prediction network for $p_\theta$ and $p_{\text{ref}}$.

By comparing the training objective functions, we can draw an interesting observation: if we denote expert samples as $\tau_w$ and generative samples as $\tau_l$, with the reference model corresponding to the generator, the sigmoid in Eq. (15) is applied to the sum of the error difference on expert and generative samples. In contrast, DPAIL evaluates these two error differences in indvidual sigmoid functions.

This distinction arises from DPO-Diffusion's derivation via the BT model, whereas DPAIL is based on the binary discriminator. Moreover, although DPO-Diffusion also aims to handle multi-modal distributions, it targets offline RL and requires preference data, setting it apart from DPAIL's focus on direct expert imitation without preference annotations.

## B  Action Execution and Diffusion Sampling in DPAIL

In DPAIL, the diffusion policy generates fixed-horizon sub-trajectories of length $H$. The resulting $H$ actions are executed sequentially in the environment, so action sequence generation is performed once every $H$ environment steps. The action execution procedure is detailed in Algorithm 2. To condition on the current state at the start of the sampling process, we overwrite the corresponding state variable at each diffusion step with the current observed state. The sampling procedure is detailed in Algorithm 3.

| **Algorithm 2** Action execution | **Algorithm 3** Sampling |
|---|---|
| 1:  $s_0 = $ env.reset() | 1:  Observe the current state $s_t$, $\tau^N \sim \mathcal{N}(\mathbf{0}, \mathbf{I})$. |
| 2:  **for** step $t = 0, 1, 2...$ **do** | 2:  **for** diffusion step $i = N, ..., 1$ **do** |
| 3:      **if** $t\%H == 0$ **then** | 3:      $z \sim \mathcal{N}(\mathbf{0}, \mathbf{I})$ |
| 4:          Sample actions $a_{0:H} \sim p_{\theta^{\text{old}}}(\cdot|s_t)$ | 4:      $\tau^{i-1} = \frac{1}{\sqrt{1-\beta_i}} \left( \tau^i - \frac{\beta_i}{\sqrt{1-\alpha_i}} \epsilon_{\theta^{\text{old}}}(\tau^i, i) \right) + \sigma_i z$ |
| 5:      **end if** | 5:      Replace the initial state in $\tau^i$ with $s_t$. |
| 6:      $a_t \leftarrow$ Get $(t\%H)$−th action in $a_{0:H}$. | 6:  **end for** |
| 7:      $r_t, s_{t+1} \leftarrow$ env.step$(a_t)$ | 7:  Get action sequence $a_{0:H}$ from $\tau^0$. |
| 8:  **end for** | |

## C  Environment Details

**HalfCheetah-v3 and Walker2d-v3**   The goal of these tasks is to move the agent forward and backward along the x-axis as quickly as possible while maintaining balance. The state includes joint angles, angular velocities and the x-coordinate. Each expert is trained using a reward function based on the the forward reward, ±x-coordinate velocity.

**Ant-v3 and AntGoal-v3**   The goal of these tasks is to control a four-legged ant robot to move forward, backward, left, or right as fast as possible while maintaining balance (Ant-v3), and to navigate to one of eight target positions evenly distributed around a circle with a radius of 20 (AntGoal-v3). The state includes joint angles, angular velocities, and the (x,y) coordinates. Each expert is trained using a reward function based on the the forward reward, ±x-coordinate velocity or ±y-coordinate velocity in Ant-v3. In AntGoal-v3, each expert is trained using a reward function based on the distance between the goal and the current robot's position. We use the 10 trajectories per mode as expert demonstrations, with each trajectory consisting of 1k transitions.

**maze2d-medium-v1 and maze2d-large-v1**   The goal of these tasks is to control a point robot to navigate to one of the target positions. In the medium-sized maze, target positions are {(1.0, 6.0), (6.0, 5.0), (6.0, 1.0)}, while in the large-sized maze, they are {(1.0, 10.0), (3.0, 8.0), (7.0, 10.0), (5.0,

4.0), (7.0, 1.0)}. Expert demonstrations are selected from D4RL dataset [1]. We use the 15 episodes for maze2d-medium-v1 and 30 episodes for maze2d-large-v1.

## D Implementation Details

**Policy gradient method** We use PPO [27] to train policies and GAE($\lambda$) to compute advantage in GAIL, DiffAIL, DRAIL and InfoGAIL. The corresponding hyperparameters for PPO are provided in Table 3. At each $k$-th iteration, we perform $m$-steps rollout in the environment. The corresponding hyperparameter settings for each algorithm are provided in Table 2.

**Diffusion and DPAIL** Both Diffusion and DPAIL utilize the same U-Net architecture with residual blocks consisting of temporal convolution and group normalization, following [17] [2]. We use $N = 50$ diffusion steps in both Diffusion and DPAIL for all tasks. Additionally, we normalize the state values before feeding them into the network. For DPAIL, we clip the norm value of $\|\epsilon - \epsilon_{\theta^{\text{old}}}(\bar{x}^i, i)\|$ not to be larger than 0.2.

**GAIL, DiffAIL, DRAIL and ASAF** For GAIL, DiffAIL, DRAIL, and ASAF, we use a multi-layer perceptron (MLP) with two hidden layers of size [64, 64] for the Gaussian policy. We also normalize the state values before feeding them into the policy network. The discriminator in GAIL is an MLP with two hidden layers of size [100, 100]. The discriminator architectures of both DiffAIL and DRAIL are based on an MLP U-Net structure based on the official repository [3], and $N = 50$ diffusion steps.

**InfoGAIL** For InfoGAIL, we use discrete latent variables, setting the number of latent variables to 8 for all tasks. We concatenate the one-hot encoding of the latent variable with the state and use the resulting vector as input to a Gaussian policy. We also normalize the state values before feeding them into the policy network. The discriminator network and class prediction network in InfoGAIL share an MLP with two hidden layers of size [100, 100] and output the corresponding values. For the coefficient of unsupervised regularization term, we perform a greedy search over the range [0.1, 0.2, 0,3, 0,5].

**Form of the reward in GAIL, DiffAIL, DRAIL and InfoGAIL** For Mujoco tasks, we use a commonly adopted reward function of the form $r(s, a) = -\log(1 - D(s, a))$, which acts as a survival bonus, encouraging agents to survive longer in the environment to accumulate more rewards. For Maze tasks, we use the reward function $r(s, a) = \log(D(s, a))$, which serves as a penalty signal. This is well-suited for goal-reaching tasks, as it incentivizes the agent to reach the goal as quickly as possible. In AntGoal-v3, we adopt the survival-style reward $r(s, a) = -\log(1 - D(s, a))$ and we find this to work well in practice.

**Details of ASAF** ASAF aims to match the trajectory distribution under a stationary policy $\pi(a|s)$. For $\pi$, the trajectory distribution $p_\pi(\tau)$ is decomposed as $p_\pi(\tau) = P(s_0) \prod_{t=0}^{T-1} \pi(a_t|s_t) P(s_{t+1}|s_t, a_t)$. To optimize Eq. (4) for trainable policy $\pi_\theta$ and generator policy $\pi_G$, ASAF defines the discriminator in policy space as

$$D_{\pi_{\theta^{\text{old}}}, \pi_\theta}(\tau) = \sigma\left(\log \frac{p_{\pi_\theta}(\tau)}{p_{\pi_{\theta^{\text{old}}}}(\tau)}\right) = \sigma\left(\sum_t \log \pi_\theta(a_t|s_t) - \log \pi_{\theta^{\text{old}}}(a_t|s_t)\right) \quad (16)$$

where the transition probability $P(s_{t+1}|s_t, a_t)$ cancels out, leaving only to the ratio of policy terms. In practice, ASAF segments trajectories into windows of length $w$, updates $\pi_\theta$ via a binary cross-entropy loss, and then sets $\pi_{\theta^{\text{old}}}$ as the updated $\pi_\theta$ to the next iteration. This procedure iteratively updates $\pi_\theta$ until convergence. We offer the ASAF algorithm in Algorithm 4.

---

[1] https://github.com/Farama-Foundation/D4RL
[2] https://github.com/jannerm/diffuser
[3] https://github.com/NVlabs/DRAIL

---

**Algorithm 4** Adversarial Soft Advantage Fitting (ASAF)

---

**Input:** expert trajectories $\mathcal{D}_E = \{\tau_n\}_{n=1}^{N_E}$
Randomly initialize $\pi_{\theta_0}$ and set $\pi_{\theta^{\text{old}}} \leftarrow \pi_{\theta_0}$
**for** $k = \{0 \cdots K\}$ **do**
  Collect trajectories $\mathcal{D}_{\theta^{\text{old}}} = \{\bar{\tau}_n\}_{n=1}^{N_{\theta^{\text{old}}}}$ using $\pi_{\theta^{\text{old}}}$ by interacting with environment
  Update $\theta_{k+1}$ by optimizing the following loss:

$$\theta_{k+1} = \arg\max_\theta \mathbb{E}_{\tau \sim \mathcal{D}_E} \left[ \log D_{\pi_{\theta^{\text{old}}}, \pi_\theta}(\tau) \right] + \mathbb{E}_{\bar{\tau} \sim \mathcal{D}_{\theta^{\text{old}}}} \left[ \log \left( 1 - D_{\pi_{\theta^{\text{old}}}, \pi_\theta}(\tau) \right) \right] ,$$

  where $D_{\pi_\theta, \pi_{\theta^{\text{old}}}}(\tau)$ is defined in Eq. (16).
  $p_{\theta^{\text{old}}} \leftarrow p_{\theta_{k+1}}$
**end for**

---

Table 2: Hyperparameters used for baselines across various environments.

| Method | Hyperparameter | HalfCheetah-v3 | Walker2d-v3 | Ant-v3 | AntGoal-v3 | maze2d-medium-v1 | maze2d-large-v1 |
|---|---|---|---|---|---|---|---|
| Diffusion | lr | 0.0002 | 0.0002 | 0.0002 | 0.0002 | 0.0002 | 0.0002 |
| | horizon $H$ | 4 | 4 | 4 | 4 | 16 | 16 |
| | # Epoch | 1000 | 1000 | 1000 | 1000 | 1000 | 1000 |
| GAIL | policy lr | 0.0003 | 0.0003 | 0.0003 | 0.0003 | 0.0003 | 0.0003 |
| | discriminator lr | 0.0003 | 0.0003 | 0.0003 | 0.0003 | 0.0003 | 0.0003 |
| | # rollout length $m$ | 50000 | 50000 | 50000 | 50000 | 10000 | 10000 |
| | # Iteration $K$ | 200 | 200 | 10000 | 600 | 100 | 100 |
| DiffAIL | policy lr | 0.0003 | 0.0003 | 0.0003 | 0.0003 | 0.0003 | 0.0003 |
| | discriminator lr | 0.0002 | 0.0002 | 0.0002 | 0.0002 | 0.0002 | 0.0002 |
| | # rollout length $m$ | 50000 | 50000 | 50000 | 50000 | 10000 | 10000 |
| | # Iteration $K$ | 200 | 200 | 10000 | 600 | 100 | 100 |
| DRAIL | policy lr | 0.0003 | 0.0003 | 0.0003 | 0.0003 | 0.0003 | 0.0003 |
| | discriminator lr | 0.0003 | 0.0003 | 0.0003 | 0.0003 | 0.0003 | 0.0003 |
| | # rollout length $m$ | 50000 | 50000 | 50000 | 50000 | 10000 | 10000 |
| | # Iteration $K$ | 200 | 200 | 10000 | 600 | 100 | 100 |
| InfoGAIL | policy lr | 0.0003 | 0.0003 | 0.0003 | 0.0003 | 0.0003 | 0.0003 |
| | discriminator lr | 0.0003 | 0.0003 | 0.0003 | 0.0003 | 0.0003 | 0.0003 |
| | coef MI | 0.2 | 0.2 | 0.1 | 0.1 | 0.3 | 0.3 |
| | # rollout length $m$ | 50000 | 50000 | 50000 | 50000 | 10000 | 10000 |
| | # Iteration $K$ | 200 | 200 | 10000 | 600 | 100 | 100 |
| ASAF | policy lr | 0.0003 | 0.0003 | 0.0003 | 0.0003 | 0.0003 | 0.0003 |
| | window $w$ | 64 | 64 | 64 | 64 | 64 | 64 |
| | # rollout length $m$ | 50000 | 50000 | 50000 | 50000 | 10000 | 10000 |
| | # Iteration $K$ | 200 | 200 | 400 | 400 | 100 | 100 |
| DPAIL (Ours) | lr | 0.0002 | 0.0002 | 0.0002 | 0.0002 | 0.0002 | 0.0002 |
| | horizon $H$ | 4 | 4 | 4 | 4 | 16 | 16 |
| | # rollout length $m$ | 10000 | 10000 | 10000 | 10000 | 5000 | 5000 |
| | # Iteration $K$ | 200 | 200 | 200 | 200 | 200 | 200 |

Table 3: PPO training hyperparameters used for each task.

| Hyperparameter | HalfCheetah-v3 | Walker2d-v3 | Ant-v3 | AntGoal-v3 | maze2d-medium-v1 | maze2d-large-v1 |
|---|---|---|---|---|---|---|
| clipping range $\epsilon$ | 0.2 | 0.2 | 0.2 | 0.2 | 0.2 | 0.2 |
| discount factor $\gamma$ | 0.99 | 0.99 | 0.99 | 0.99 | 0.995 | 0.995 |
| gae parameter $\lambda$ | 0.97 | 0.97 | 0.97 | 0.97 | 0.97 | 0.97 |
| # epoch per iteration | 50 | 50 | 50 | 50 | 40 | 40 |

# E    Additional Experimental Results

We present the learned behaviors of baselines methods and DPAIL in Figures 6 and 7. The expert demonstration behaviors are visualized in Figure 5. To evaluate the multi-modal learning capability of imitation learning methods, we measure the entropy of the learned behaviors to quantify diversity. Additionally, to assess the similarity between the learned trajectories and expert demonstrations, we compute Maximum Mean Discrepancy (MMD) between their respective state-action distributions, using an RBF kernel with 20 bandwidths, as shown in Table 4. Since MMD quantifies the divergence between distributions, lower values indicate better recovery of all modes present in the expert distribution.

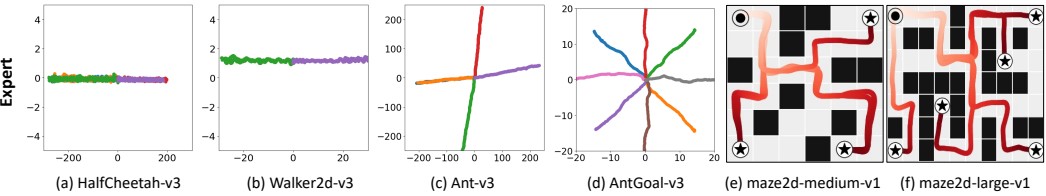

Figure 5: Expert demonstrations across 6 tasks.

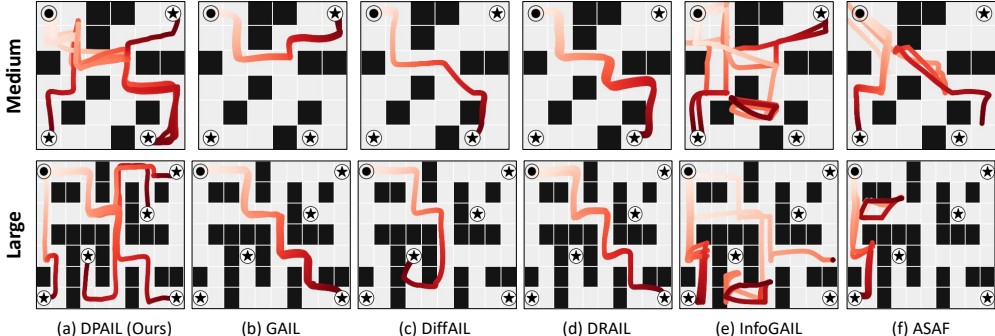

Figure 6: Learned behaviors of baseline methods and DPAIL (ours) in Maze2d tasks. The first row depicts maze2d-medium-v1, while the second row depicts maze2d-large-v1. Each graph illustrates 5 different trajectories generated by the same policy. The initial position is marked with circle, and the goal positions are marked with stars.

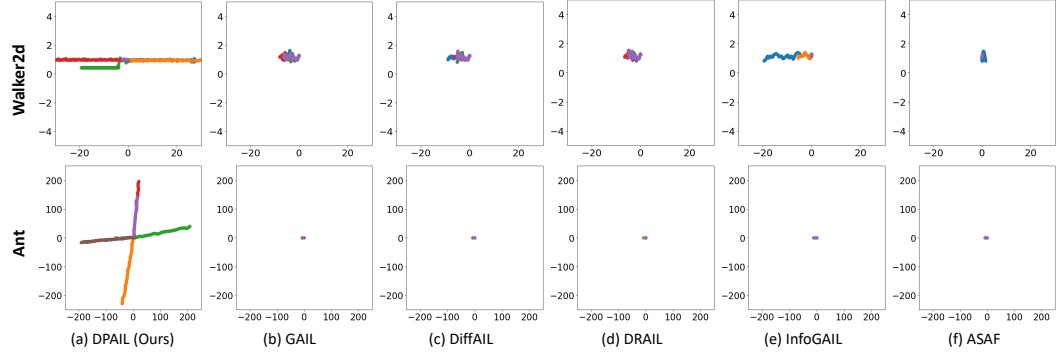

Figure 7: Learned behaviors of baseline methods and DPAIL (ours) in MuJoCo tasks. The first row shows Walker2d-v3, where the task is to move forward and backward (±x-axis). The second row shows Ant-v3, where the task is to move forward, backward, left and right (±x-axis, ±y-axis). Each plot illustrates ten different trajectories generated by the same policy.

## E.1    Impact of the Number of Demonstrations and Modes

We provide additional experimental results on the impact of the number of demonstrations and modes in Ant-v3 (Table 5) and AntGoal-v3 (Table 6). In both tasks, increasing the number of modes generally degrades the performance of most algorithms. However, DPAIL exhibits greater robustness, benefiting from the expressiveness of diffusion models.

Table 4: MMD($\downarrow$) between state-action distributions between expert demonstrations and learned behaviors. DPAIL has the lowest value on most tasks, indicating better recovery of expert distributions.

| Environment | BC | Diffusion | GAIL | DiffAIL | DRAIL | InfoGAIL | ASAF | DPAIL |
|---|---|---|---|---|---|---|---|---|
| HalfCheetah-v3 | 0.029 | 0.027 | 0.038 | 0.038 | 0.045 | 0.039 | 0.031 | **0.025** |
| Walker2d-v3 | 0.127 | **0.091** | 0.165 | 0.182 | 0.212 | 0.099 | 0.190 | 0.096 |
| Ant-v3 | 0.334 | 0.018 | 0.281 | 0.341 | 0.335 | 0.282 | 0.311 | **0.012** |
| AntGoal-v3 | 0.571 | 0.082 | 0.385 | 0.443 | 0.401 | 0.192 | 0.551 | **0.021** |
| maze2d-medium-v1 | 4.9e-4 | 9.1e-5 | 4.8e-4 | 4.2e-4 | 4.5e-4 | 5.0e-4 | 5.5e-4 | **8.0e-5** |
| maze2d-large-v1 | 5.1e-3 | 1.0e-4 | 3.0e-4 | 1.5e-4 | 5.0e-4 | 5.3e-4 | 5.9e-3 | **9.2e-5** |

Table 5: Normalized score (Score) and entropy (Ent) on varying the number of demonstrations and modes in Ant-v3.

| # of modes | # of demos | metrics | BC | Diffusion | GAIL | DiffAIL | DRAIL | InfoGAIL | ASAF | DPAIL |
|---|---|---|---|---|---|---|---|---|---|---|
| 1 | 3 | Score | 0.13±0.03 | 0.38±0.12 | 0.39±0.08 | **0.50**±0.20 | 0.27±0.12 | 0.23±0.03 | 0.27±0.12 | 0.48±0.25 |
| | | Ent | 0.00±0.00 | 0.00±0.00 | 0.00±0.00 | 0.00±0.00 | 0.00±0.00 | 0.00±0.00 | 0.00±0.00 | 0.00±0.00 |
| | 5 | Score | 0.15±0.06 | 0.4±0.23 | 0.39±0.08 | **0.55**±0.25 | 0.22±0.15 | 0.21±0.01 | 0.22±0.15 | 0.53±0.22 |
| | | Ent | 0.00±0.00 | 0.00±0.00 | 0.00±0.00 | 0.00±0.00 | 0.00±0.00 | 0.00±0.00 | 0.00±0.00 | 0.00±0.00 |
| | 10 | Score | 0.17±0.09 | 0.57±0.21 | 0.50±0.18 | 0.55±0.40 | 0.47±0.28 | 0.23±0.00 | 0.47±0.28 | **0.67**±0.29 |
| | | Ent | 0.00±0.00 | 0.00±0.00 | 0.00±0.00 | 0.00±0.00 | 0.00±0.00 | 0.00±0.00 | 0.00±0.00 | 0.00±0.00 |
| | 20 | Score | 0.10±0.09 | 0.74±0.08 | 0.51±0.12 | 0.69±0.14 | 0.45±0.24 | 0.25±0.07 | 0.45±0.24 | **0.72**±0.05 |
| | | Ent | 0.00±0.00 | 0.00±0.00 | 0.00±0.00 | 0.00±0.00 | 0.00±0.00 | 0.00±0.00 | 0.00±0.00 | 0.00±0.00 |
| 2 | 3 | Score | 0.10±0.06 | 0.40±0.22 | 0.17±0.19 | 0.14±0.29 | 0.28±0.10 | 0.12±0.01 | 0.10±0.12 | **0.47**±0.10 |
| | | Ent | 0.75±0.30 | 0.60±0.08 | 0.36±0.20 | 0.48±0.18 | 0.29±0.18 | 0.52±0.22 | 0.42±0.29 | 0.62±0.11 |
| | 5 | Score | 0.14±0.06 | 0.43±0.15 | 0.17±0.25 | 0.13±0.25 | 0.26±0.10 | 0.15±0.01 | 0.20±0.14 | **0.48**±0.08 |
| | | Ent | 0.88±0.09 | 0.49±0.24 | 0.34±0.18 | 0.46±0.26 | 0.31±0.18 | 0.53±0.26 | 0.46±0.32 | 0.60±0.13 |
| | 10 | Score | 0.18±0.15 | 0.52±0.09 | 0.15±0.23 | 0.26±0.29 | 0.31±0.23 | 0.13±0.02 | 0.15±0.02 | **0.64**±0.08 |
| | | Ent | 0.71±0.41 | 0.59±0.09 | 0.18±0.15 | 0.52±0.30 | 0.19±0.15 | 0.65±0.15 | 0.39±0.25 | 0.61±0.04 |
| | 20 | Score | 0.15±0.03 | 0.59±0.09 | 0.16±0.02 | 0.32±0.25 | 0.28±0.11 | 0.12±0.01 | 0.12±0.09 | **0.63**±0.18 |
| | | Ent | 1.01±0.25 | 0.66±0.01 | 0.44±0.25 | 0.51±0.23 | 0.36±0.32 | 0.52±0.18 | 0.44±0.30 | 0.59±0.08 |
| 3 | 3 | Score | 0.05±0.01 | 0.34±0.11 | 0.13±0.20 | 0.12±0.01 | 0.08±0.08 | 0.03±0.01 | 0.03±0.01 | **0.53**±0.07 |
| | | Ent | 0.93±0.32 | 0.85±0.11 | 0.76±0.17 | 0.82±0.19 | 0.98±0.10 | 0.95±0.08 | 0.81±0.12 | 1.01±0.16 |
| | 5 | Score | 0.06±0.05 | 0.44±0.13 | 0.11±0.21 | 0.10±0.24 | 0.10±0.05 | 0.02±0.01 | 0.01±0.02 | **0.55**±0.03 |
| | | Ent | 0.90±0.46 | 0.81±0.16 | 0.73±0.41 | 0.68±0.17 | 0.57±0.30 | 0.93±0.12 | 0.79±0.10 | 0.62±0.18 |
| | 10 | Score | 0.06±0.01 | 0.43±0.17 | 0.12±0.11 | 0.13±0.13 | 0.07±0.05 | 0.02±0.01 | 0.01±0.01 | **0.55**±0.13 |
| | | Ent | 0.90±0.29 | 0.91±0.16 | 0.82±0.18 | 0.90±0.07 | 0.69±0.37 | 0.99±0.08 | 0.75±0.38 | 1.11±0.50 |
| | 20 | Score | 0.03±0.04 | 0.60±0.04 | 0.11±0.13 | 0.15±0.05 | 0.12±0.09 | 0.01±0.00 | 0.05±0.06 | **0.64**±0.14 |
| | | Ent | 1.00±0.24 | 0.88±0.06 | 0.83±0.14 | 0.96±0.12 | 0.98±0.09 | 1.06±0.03 | 0.65±0.48 | 0.91±0.14 |
| 4 | 3 | Score | 0.08±0.01 | 0.21±0.11 | 0.08±0.03 | 0.02±0.01 | 0.02±0.01 | 0.03±0.01 | 0.00±0.00 | **0.39**±0.06 |
| | | Ent | 0.90±0.27 | 1.18±0.05 | 0.85±0.42 | 1.22±0.08 | 1.06±0.05 | 1.07±0.08 | 1.01±0.19 | 1.20±0.11 |
| | 5 | Score | 0.09±0.01 | 0.28±0.09 | 0.01±0.00 | 0.04±0.05 | 0.07±0.09 | 0.02±0.01 | 0.01±0.00 | **0.42**±0.01 |
| | | Ent | 0.95±0.21 | 1.17±0.06 | 0.95±0.09 | 1.31±0.04 | 1.07±0.70 | 1.19±0.08 | 0.89±0.16 | 1.20±0.14 |
| | 10 | Score | 0.06±0.07 | 0.48±0.11 | 0.01±0.00 | 0.02±0.01 | 0.02±0.08 | 0.03±0.02 | 0.00±0.00 | **0.56**±0.06 |
| | | Ent | 0.75±0.22 | 1.22±0.12 | 1.19±0.10 | 1.21±0.10 | 1.26±0.11 | 1.04±0.07 | 1.07±0.19 | 1.21±0.05 |
| | 20 | Score | 0.05±0.05 | 0.60±0.02 | 0.01±0.00 | 0.04±0.02 | 0.03±0.01 | 0.02±0.02 | 0.00±0.00 | **0.65**±0.04 |
| | | Ent | 1.01±0.36 | 1.13±0.08 | 1.29±0.03 | 1.31±0.06 | 1.30±0.05 | 1.31±0.05 | 0.87±0.35 | 1.18±0.15 |

Table 6: Normalized score (Score) and entropy (Ent) on varying the number of demonstrations and modes in AntGoal-v3.

| # of modes | # of demos | metrics | BC | Diffusion | GAIL | DiffAIL | DRAIL | InfoGAIL | ASAF | DPAIL |
|---|---|---|---|---|---|---|---|---|---|---|
| 1 | 3 | Score | 0.39±0.17 | 0.62±0.18 | 0.74±0.08 | **0.94**±0.02 | 0.78±0.10 | 0.62±0.04 | 0.40±0.21 | 0.80±0.02 |
| | | Ent | 0.00±0.00 | 0.00±0.00 | 0.00±0.00 | 0.00±0.00 | 0.00±0.00 | 0.00±0.00 | 0.00±0.00 | 0.00±0.00 |
| | 5 | Score | 0.43±0.20 | 0.65±0.21 | 0.84±0.09 | **0.96**±0.02 | 0.83±0.05 | 0.68±0.09 | 0.41±0.38 | 0.83±0.08 |
| | | Ent | 0.00±0.00 | 0.00±0.00 | 0.00±0.00 | 0.00±0.00 | 0.00±0.00 | 0.00±0.00 | 0.00±0.00 | 0.00±0.00 |
| | 10 | Score | 0.46±0.14 | 0.79±0.05 | 0.92±0.02 | **0.94**±0.02 | 0.93±0.04 | 0.64±0.13 | 0.47±0.26 | 0.82±0.12 |
| | | Ent | 0.00±0.00 | 0.00±0.00 | 0.00±0.00 | 0.00±0.00 | 0.00±0.00 | 0.00±0.00 | 0.00±0.00 | 0.00±0.00 |
| | 20 | Score | 0.52±0.20 | 0.84±0.04 | 0.86±0.05 | **0.94**±0.01 | 0.80±0.07 | 0.65±0.00 | 0.53±0.16 | 0.88±0.01 |
| | | Ent | 0.00±0.00 | 0.00±0.00 | 0.00±0.00 | 0.00±0.00 | 0.00±0.00 | 0.00±0.00 | 0.00±0.00 | 0.00±0.00 |
| 2 | 3 | Score | 0.08±0.03 | 0.51±0.08 | 0.68±0.22 | **0.81**±0.04 | 0.72±0.07 | 0.45±0.06 | 0.02±0.01 | 0.69±0.10 |
| | | Ent | 0.50±0.13 | 0.46±0.23 | 0.20±0.17 | 0.54±0.09 | 0.23±0.10 | 0.54±0.21 | 0.51±0.27 | 0.68±0.12 |
| | 5 | Score | 0.10±0.08 | 0.54±0.19 | 0.74±0.11 | **0.86**±0.08 | 0.74±0.13 | 0.47±0.10 | 0.04±0.04 | 0.71±0.07 |
| | | Ent | 0.56±0.12 | 0.42±0.33 | 0.15±0.11 | 0.64±0.05 | 0.20±0.09 | 0.64±0.17 | 0.53±0.31 | 0.71±0.17 |
| | 10 | Score | 0.13±0.09 | 0.58±0.21 | 0.77±0.07 | 0.80±0.07 | 0.72±0.05 | 0.49±0.12 | 0.03±0.03 | **0.80**±0.06 |
| | | Ent | 0.68±0.10 | 0.42±0.24 | 0.17±0.17 | 0.53±0.14 | 0.15±0.13 | 0.68±0.08 | 0.65±0.25 | 0.52±0.18 |
| | 20 | Score | 0.15±0.09 | 0.83±0.09 | 0.75±0.11 | 0.86±0.07 | 0.76±0.13 | 0.49±0.07 | 0.16±0.15 | **0.87**±0.04 |
| | | Ent | 0.54±0.13 | 0.33±0.18 | 0.43±0.18 | 0.43±0.18 | 0.43±0.16 | 0.61±0.16 | 0.51±0.17 | 0.52±0.15 |
| 4 | 3 | Score | 0.01±0.00 | 0.37±0.12 | 0.52±0.13 | 0.52±0.21 | 0.54±0.22 | 0.40±0.28 | 0.02±0.01 | **0.64**±0.10 |
| | | Ent | 0.94±0.05 | 0.99±0.13 | 1.02±0.28 | 1.02±0.09 | 1.04±0.20 | 1.19±0.16 | 0.92±0.28 | 1.14±0.26 |
| | 5 | Score | 0.03±0.01 | 0.39±0.26 | 0.54±0.09 | 0.50±0.19 | 0.50±0.17 | 0.38±0.11 | 0.03±0.04 | **0.66**±0.07 |
| | | Ent | 0.97±0.03 | 1.00±0.15 | 1.00±0.38 | 1.09±0.18 | 1.05±0.18 | 1.23±0.15 | 0.82±0.30 | 1.16±0.27 |
| | 10 | Score | 0.04±0.02 | 0.45±0.28 | 0.67±0.95 | 0.66±0.08 | 0.67±0.11 | 0.52±0.14 | 0.15±0.31 | **0.75**±0.06 |
| | | Ent | 1.02±0.07 | 1.05±0.06 | 0.88±0.21 | 0.94±0.24 | 0.96±0.25 | 1.25±0.12 | 0.56±0.42 | 1.05±0.17 |
| | 20 | Score | 0.02±0.01 | 0.39±0.26 | 0.54±0.09 | 0.50±0.19 | 0.50±0.17 | 0.38±0.11 | 0.03±0.04 | **0.66**±0.07 |
| | | Ent | 0.98±0.26 | 1.00±0.15 | 1.00±0.38 | 1.09±0.18 | 1.05±0.18 | 1.23±0.15 | 0.82±0.30 | 1.16±0.27 |
| 8 | 3 | Score | 0.02±0.01 | 0.20±0.07 | 0.43±0.11 | 0.32±0.04 | 0.39±0.07 | 0.39±0.11 | 0.01±0.00 | **0.52**±0.05 |
| | | Ent | 1.26±0.10 | 1.50±0.38 | 1.48±0.27 | 1.78±0.12 | 1.79±0.23 | 1.77±0.05 | 1.43±0.33 | 1.70±0.27 |
| | 5 | Score | 0.03±0.01 | 0.23±0.09 | 0.47±0.06 | 0.37±0.08 | 0.40±0.10 | 0.45±0.09 | 0.01±0.00 | **0.54**±0.03 |
| | | Ent | 0.98±0.20 | 1.56±0.48 | 1.78±0.17 | 1.82±0.13 | 1.81±0.13 | 1.79±0.08 | 1.65±0.23 | 1.76±0.41 |
| | 10 | Score | 0.04±0.04 | 0.22±0.07 | 0.58±0.05 | 0.35±0.16 | 0.41±0.15 | 0.48±0.07 | 0.01±0.00 | **0.67**±0.03 |
| | | Ent | 1.46±0.58 | 1.52±0.31 | 1.51±0.17 | 1.75±0.11 | 1.72±0.07 | 1.78±0.13 | 1.52±0.27 | 1.73±0.23 |
| | 20 | Score | 0.01±0.00 | 0.53±0.18 | 0.58±0.02 | 0.45±0.08 | 0.41±0.10 | 0.46±0.02 | 0.01±0.00 | **0.74**±0.02 |
| | | Ent | 1.51±0.24 | 1.18±0.41 | 1.30±0.14 | 1.75±0.20 | 1.75±0.23 | 1.79±0.12 | 1.46±0.16 | 1.78±0.44 |

# F Derivation with latent variable $z$

We define conditional diffusion models with an latent variable $z$, where the joint probability $p_\theta(\tau^{0:N}, z)$ factorizes as:

$$p_\theta(\tau^{0:N}, z) = q(\tau^N)q(z)\prod_{t=1}^{N} p_\theta(\tau^{i-1}|\tau^i, z)$$

where $q(z)$ is the prior and $q(\tau^N)$ is the standard normal distribution. To evaluate the log-ratio $\log \frac{p_\theta(\tau^0)}{p_{\theta^{old}}(\tau^0)}$, we rewrite it with the posterior $q(z|\tau^0)$ as:

$$\log \frac{p_\theta(\tau^0)}{p_{\theta^{old}}(\tau^0)} = \frac{\prod_{i=1}^{N} q(\tau^i|\tau^{i-1})}{\prod_{i=1}^{N} q(\tau^i|\tau^{i-1})} \frac{q(z|\tau^0)}{q(z|\tau^0)} \frac{p_\theta(\tau^0)}{p_{\theta^{old}}(\tau^0)} = \frac{\prod_{i=1}^{N} p_\theta(\tau^{i-1}|\tau^i, z)}{\prod_{i=1}^{N} p_{\theta^{old}}(\tau^{i-1}|\tau^i, z)} \frac{q(z)}{q(z)} \frac{q(\tau^N)}{q(\tau^N)}$$

where $q(\tau^N)$ and $q(z)$ cancel out. Taking expectation of this log-ratio over $\tau^{1:N} \sim q(\cdot|\tau^0)$, $z \sim q(\cdot|\tau^0)$ does not change the value.

$$\max_\theta \mathbb{E}_{\tau^0 \sim p_E}\left[\log \sigma\left(\mathbb{E}_{\substack{z \sim q_\phi(z|\tau^0),\\ \tau^{1:N} \sim q(\tau^{1:N}|\tau^0)}} \log \frac{\prod_{i=1}^{N} p_\theta(\tau^{i-1}|\tau^i, z)}{\prod_{i=1}^{N} p_{\theta^{old}}(\tau^{i-1}|\tau^i, z)}\right)\right]$$

Given the expert sample $\tau^0 \sim p_E$, the approximate variational posterior $q_\phi(z|\tau^0)$, the reverse process $p_\theta(\tau^{i-1}|\tau^i, z)$, and the corresponding forward process $q(\tau^{1:N}|\tau^0)$, these lead to the following lowerbound:

$$\mathbb{E}_{\tau^0 \sim p_E}\left[\log \sigma\left(\mathbb{E}_{\substack{z \sim q_\phi(z|\tau^0)\\ \tau^{1:N} \sim q(\tau^{1:N}|\tau^0)}} \log \frac{\prod_{i=1}^{N} p_\theta(\tau^{i-1}|\tau^i, z)}{\prod_{i=1}^{N} p_{\theta^{old}}(\tau^{i-1}|\tau^i, z)}\right)\right]$$

$$= \mathbb{E}_{\tau^0 \sim p_E}\left[\log \sigma\left(\mathbb{E}_{\substack{z \sim q_\phi(z|\tau^0)\\ \tau^{1:N} \sim q(\tau^{1:N}|\tau^0)}} \sum_{i=1}^{N} \log \frac{p_\theta(\tau^{i-1}|\tau^i, z)}{p_{\theta^{old}}(\tau^{i-1}|\tau^i, z)}\right)\right]$$

$$= \mathbb{E}_{\tau^0 \sim p_E}\left[\log \sigma\left(\sum_{i=1}^{N} \mathbb{E}_{\substack{z \sim q_\phi(z|\tau^0)\\ \tau^{1:N} \sim q(\tau^{1:N}|\tau^0)}} \log \frac{p_\theta(\tau^{i-1}|\tau^i, z)}{p_{\theta^{old}}(\tau^{i-1}|\tau^i, z)}\right)\right]$$

$$= \mathbb{E}_{\tau^0 \sim p_E}\left[\log \sigma\left(\sum_{i=1}^{N} \mathbb{E}_{\substack{z \sim q_\phi(z|\tau^0)\\ \tau^{i-1},\tau^i \sim q(\tau^i|\tau^0)q(\tau^{i-1}|\tau^i,\tau^0)}} \log \frac{p_\theta(\tau^{i-1}|\tau^i, z)}{p_{\theta^{old}}(\tau^{i-1}|\tau^i, z)}\right)\right]$$

$$\geq \mathbb{E}_{\tau^0 \sim p_E}\left[\mathbb{E}_{\substack{i \sim \mathcal{U}(1,N),\\ z \sim q_\phi(z|\tau^0),\tau^i \sim q(\tau^i|\tau^0)}} \log \sigma\left(N\mathbb{E}_{\tau^{i-1} \sim q(\tau^{i-1}|\tau^i,\tau^0)} \log \frac{p_\theta(\tau^{i-1}|\tau^i, z)}{p_{\theta^{old}}(\tau^{i-1}|\tau^i, z)}\right)\right] \quad \text{(concavity)}$$

$$= \mathbb{E}_{\tau^0 \sim p_E}\left[\mathbb{E}_{\substack{i \sim \mathcal{U}(1,N),\\ z \sim q_\phi(z|\tau^0),\tau^i \sim q(\tau^i|\tau^0)}} \log \sigma\left(N\mathbb{E}_{\tau^{i-1} \sim q(\tau^{i-1}|\tau^i,\tau^0)} \log \frac{q(\tau^{i-1}|\tau^i, \tau^0)}{p_{\theta^{old}}(\tau^{i-1}|\tau^i, z)} - \log \frac{q(\tau^{i-1}|\tau^i, \tau^0)}{p_\theta(\tau^{i-1}|\tau^i, z)}\right)\right]$$

$$\therefore \mathcal{L}_{DPAIL}^{(1)}(\theta, \theta^{old}, \tau^0, z) = \mathbb{E}_{\tau^0, z, i, \epsilon}\left[\log \sigma\left(N \cdot C_i\left(\|\epsilon - \epsilon_{\theta^{old}}(\tau^i, z, i)\|^2 - \|\epsilon - \epsilon_\theta(\tau^i, z, i)\|^2\right)\right)\right].$$

Here, $C_i = \frac{\beta_i}{(1-\beta_i)(1-\alpha_i)}$, $z \sim q_\phi(z|\tau^0)$, $\epsilon \sim \mathcal{N}(0, I)$, and $\tau^i \sim q(\tau^i|\tau^0)$, thus $\tau^i = \sqrt{\alpha_i}\tau^0 + (1-\alpha_i)\epsilon$. Similarly, for given generative samples $\bar{\tau} \sim p_{\theta^{old}}$, we can also get $\mathcal{L}_{DPAIL}^{(2)}(\theta, \theta^{old}, \bar{\tau}^0, z)$. The overall objective becomes:

$$\therefore \mathbb{E}_{\substack{\tau^0 \sim p_E,\\ z \sim q_\phi(z|\tau^0)}}\left[\mathcal{L}_{DPAIL}^{(1)}(\theta, \theta^{old}, \tau^0, z)\right] + \mathbb{E}_{\substack{\bar{\tau}^0 \sim p_{\theta^{old}},\\ z \sim q_\phi(z|\bar{\tau}^0)}}\left[\mathcal{L}_{DPAIL}^{(2)}(\theta, \theta^{old}, \bar{\tau}^0, z)\right] + \underbrace{\left[\mathbb{E}_{q_\phi(\tau^0, z)}[\log \frac{q_\phi(\tau^0, z)}{q(\tau^0)q_\phi(z)}]\right]}_{:=\text{MI}(\tau_0, z)}$$

where the last term encourages learning meaningful unsupervised representation $z \sim q_\phi(z|\bar{\tau})$ by maximizing mutual information.

