# OpenReview forum: "DPAIL: Training Diffusion Policy for Adversarial Imitation Learning without Policy Optimization"
_NeurIPS.cc/2025/Conference — NeurIPS 2025 poster_

### Official Review · Reviewer_hTNL · 2025-06-30

**Clarity:** 3
**Significance:** 3
**Originality:** 3
**Rating:** 4
**Confidence:** 4

**Summary:**

This paper introduces DPAIL, an adversarial Imitation Learning (AIL) framework that leverages diffusion models as a policy class to enhance expressiveness. Building upon the Adversarial Soft Advantage Fitting (ASAF) framework, which reformulates AIL to remove explicit policy gradients, the authors address the challenge that exact likelihood evaluation is intractable for diffusion models. To overcome this, they derive a tractable lower bound for the discriminator objective, making it feasible to optimize the diffusion policy within the ASAF paradigm. The resulting approach retains the expressiveness of diffusion models while ensuring the stability and efficiency of adversarial training. Extensive experiments on continuous control (MuJoCo) and navigation (Maze2D) tasks demonstrate that DPAIL can successfully imitate diverse expert strategies, maintaining high task performance and behavioral diversity, especially in environments with multiple behavior modes where traditional methods tend to collapse to a single mode.

**Questions:**

1.Could the authors clarify how hyperparameters for all baselines (e.g., GAIL, DiffAIL, InfoGAIL) were tuned? Were the same budgets and random seeds applied?
2. If the same tuning effort was not feasible, what measures were taken to ensure a fair comparison?
3. Do the authors have any preliminary ideas or results on integrating latent variables or guided diffusion for mode conditioning? Could such an extension be compatible with the current ASAF lower bound design?

**Ethical Concerns:**

["NO or VERY MINOR ethics concerns only"]

**Final Justification:**

After considering the rebuttal, author clarifications, and discussions with other reviewers , my recommendation remains Borderline Accept, with the following reasoning:

Resolved Issues：
(1) Differences from DRAIL / DiffAIL – The authors provided a clear and convincing explanation of the fundamental differences, particularly the implicit discriminator formulation in DPAIL, which justifies its ability to capture multi-modal behaviors.
(2) Hyperparameter Tuning Fairness – Detailed descriptions of the tuning process for baselines (including rollout lengths, iteration counts, and seeds) addressed concerns about fairness in comparisons.
(3) Sensitivity to Additional Hyperparameter – Additional experiments showed performance trends with varying generated sample counts, clarifying that the method is reasonably robust once above a threshold.

Remaining Concerns:
(1) Inference Latency and Real-World Scalability – The method’s reliance on diffusion policies still results in noticeably higher inference time, with no concrete acceleration strategies implemented; this limits immediate deployment potential.
(2) Evaluation Scope – Experiments remain focused on simulated environments; real-world or noisy/heterogeneous scenarios are not tested, leaving generalizability uncertain.

I assign strong weight to the technical novelty and solid methodological grounding, as the rebuttal clarified key differentiators and experimental fairness. However, the limited evaluation scope and scalability concerns are significant practical drawbacks, preventing a full accept. Overall, the paper is technically sound and makes a meaningful contribution, but the above limitations temper my enthusiasm, leading to a borderline accept recommendation.

**Limitations:**

The authors have explicitly acknowledged the main limitations of their work in the dedicated Limitations section. They correctly point out that:
(1) The iterative sampling cost of diffusion policies introduces significant computational overhead at inference time compared to simpler policy classes.
(2) The method currently lacks latent controllability, making it harder to guide the imitation policy towards a user-desired behavior mode compared to methods like InfoGAIL.

**Paper Formatting Concerns:**

The references in the main text are not cited in strictly ascending numerical order. For consistency and readability, please ensure that all references are cited sequentially in the order of their appearance in the text.

**Quality:**

3

**Strengths And Weaknesses:**

Strengths：
1. The paper is technically solid overall, providing a well-motivated derivation of a tractable lower bound for training diffusion-based policies under the ASAF framework.
2. Experimental results are extensive and conducted on a variety of benchmark tasks (MuJoCo, Maze2D) with different levels of multi-modal behaviors, and the metrics include both normalized returns and entropy for diversity, which are appropriate and relevant. The implementation details are carefully discussed, and the paper provides reproducibility information, e.g., hyperparameters, baseline setups.
3. The method addresses a practical challenge in imitation learning—efficiently learning from diverse demonstrations without costly policy gradients—an issue of broad interest in the field. Combining the strengths of AIL (robust generalization) and diffusion models (multi-modal expressivity) can have significant implications for robotics and other sequential decision-making tasks.

Weaknesses：
1.The method’s practical scalability and applicability in real-world scenarios with strict latency constraints remain somewhat unclear.
2. Hyperparameter sensitivity is only partially discussed.
3.While the integration of ASAF with diffusion models is novel, parts of the loss design (e.g., using noise prediction errors) share structural similarities with prior works like DRAIL and DiffAIL. The paper could strengthen its argument about how its approach is fundamentally different.

---

> ### Author Rebuttal · Authors · 2025-07-31
>
> **(Weakness 1) Inference latency and reward-world applicability**
>
> As noted in the section on limitations, DPAIL incurs longer inference time due to the use of diffusion models as policies. This is a trade-off for their high expressiveness and ability to capture multi-modal behaviors, which are essential for our goal of modeling diverse expert demonstrations.
> There has been promising progress in accelerating diffusion model inference (e.g., through distillation or fast samplers), and integrating such techniques into our framework is a promising direction for improving scalability and real-time applicability, which we leave as future work.
>
> **(Weakness 2) Sensitivity of the additional hyperparameter**
>
> In addition to the hyperparameter planning horizon $H$ and sampling step $N$ discussed in the experiments section, we conducted additional sensitivity analysis on the number of generated samples per iteration in AntGoal-v3 environment.
> The results are shown below:
>
> |# of generated samples per iteration | 0 | 1000 | 5000 | 10000 | 20000 |
> |-|-|-|-|-|-|
> |Normalized Score| 0.43 | 0.51 | 0.65 | 0.67 | 0.68 |
>
> Our findings indicate that when the number of generated samples is too small, performance degrades due to insufficient learning from the generated samples. However, once the sample size exceeds a certain threshold, performance improves and then saturates, exhibiting only marginal gains with further increases.
>
> **(Weakness 3) Differences from DRAIL and DiffAIL**
>
> While DPAIL shares similarities with prior works such as DRAIL and DiffAIL, the underlying mechanisms are fundamentally different.
> Both DRAIL and DiffAIL use explicit diffusion-based discriminators, and then used to provide rewards for training a separate, uni-modal Gaussian policy, often via RL.
> In contrast, DPAIL does not use an explicit discriminator at all. Instead, the discriminator is implicitly defined by the diffusion policy itself through the ASAF framework. This implicit formulation enables DPAIL to capture and reproduce diverse behaviors from multi-modal demonstrations, whereas DRAIL and DiffAIL are limited in this regard since their diffusion models are used solely as a reward signal for training unimodal policies.
>
> **(Question 1 & 2) Hyperparameter Tuning**
>
> We made efforts to ensure a fair and meaningful comparison across all baseline algorithms.
> For PPO-based baselines (e.g., GAIL, DiffAIL, InfoGAIL), we performed hyperparameter tuning on several factors that impact performance.
> For example, we searched over the # of rollout-length per iteration, using values [5e3, 1e4, 5e4, 1e5]. In our experiments on MuJoCo tasks, rollout lengths of $\geq 5 \times 10^4$ generally led to stable and successful training, so we selected those as default settings for the PPO-based baselines.
>
> Another hyperparameter is K, the # of iterations. Since different algorithms require different numbers of iterations $K$ to reach their converged performance, we determined a sufficiently large value of $K$ for each method and task during hyperparameter tuning.
> This ensures that performance is evaluated after sufficiently training rather than at a fixed iteration cutoff.
> While our method requires fewer environment interactions to converge, we allowed other baselines to consume more interactions if needed to reach their converged performance.
> This ensures a fair comparion based on fully trained policies, rather than under a fixed interaction budget.
> All experiments were run using the same set of random seeds (0-4). For details about the tuned hyperparameters, please refer to Appendix D (Implementation Details).
>
> **(Question 3) Extension to latent variables and Conditional Diffusion Policies**
>
> Yes, we have preliminary ideas on integrating latent variables $z$ for mode conditioning, and such an extension is compatible with the current DPAIL lower bound. Below, we briefly outline the idea.
> We define a conditional diffusion model with a latent variable $z$, where the joint probability $p\_\theta(\tau^{0:N}, z)$ factorizes as:
>
> $p\_\theta(\tau^{0:N}, z) = q(\tau^N)q(z) \prod\_{i=1}^{N} p\_\theta(\tau^{i-1} | \tau^i, z)$.
>
> where $q(z)$ is the prior of $z$, $q(\tau^N)$ is a standard normal distribution.
>
> To evaluate the log-ratio $\log \frac{p\_\theta(\tau^0)}{p\_{\theta^\text{old}}(\tau^0)}$,
> we rewrite it with the posterior $q(z|\tau^0)$ as:
>
> $\log \frac{p\_\theta(\tau^0)}{p\_{\theta^\text{old}}(\tau^0)}
> = \log \frac{\prod\_{i=1}^N q(\tau^i|\tau^{i-1})}{\prod\_{i=1}^N q(\tau^i | \tau^{i-1})}
> \frac{q(z|\tau^0)}{q(z|\tau^0)}
> \frac{p\_\theta(\tau^0)}{p\_{\theta^\text{old}}(\tau^0)}
> =\log \frac{\prod\_{i=1}^N p\_\theta(\tau^{i-1}|\tau^{i}, z)}{\prod\_{i=1}^N p\_{\theta^{\text{old}}}(\tau^{i-1} | \tau^{i}, z)}
> \frac{q(z)}{q(z)}
> \frac{q(\tau^N)}{q(\tau^N)}
> $,
>
> where $q(\tau^N)$ and $q(z)$ cancel out.
> Taking expectation of this log-ratio over $\tau^{1:N} \sim q(\cdot | \tau^0), z~\sim q(z|\tau^0)$ does not change the value since it is still a ratio of marginal. Thus, the objective can be rewritten as:
>
> $\max\_\theta \mathbb{E}\_{\tau^0 \sim p_E} \left[
> \log \sigma \left (
> \mathbb{E}\_{\substack{z \sim q\_(z | \tau^0),\\\\ \tau^{1:N} \sim q(\tau^{1:N} | \tau^0)}}
> \log \frac{\prod\_{i=1}^N p\_\theta(\tau^{i-1}|\tau^{i}, z)}
> {\prod\_{i=1}^N p\_{\theta_{\text{old}}}(\tau^{i-1}|\tau^{i}, z)}
> \right) \right].$
>
> Given the expert sample $\tau^0\sim p_E$, the approximate variational posterior $q_\phi(z|\tau^0)$, the reverse process $p_\theta(\tau^{i-1}| \tau^i, z)$, this leads to the following lowerbound:
>
> $\begin{align*} &\mathbb{E}\_{\tau^{0} \sim p\_{E}}
> \left [ \log \sigma \left(
> \mathbb{E}\_{\substack{z \sim q\_\phi(z|\tau^0) \\\\ {\tau^{1:N} \sim q(\tau^{1:N}| \tau^0)}}}
> \log \frac{\prod\_{i=1}^{N} p\_\theta(\tau^{i-1}|\tau^{i},z)}
> {\prod\_{i=1}^{N} p\_{\theta^{\text{old}}}(\tau^{i-1}|\tau^{i},z)}
> \right) \right] \\\\
> &= \mathbb{E}\_{\tau^{0} \sim p\_{E}}
> \left [ \log \sigma \left(
> \mathbb{E}\_{\substack{z\sim q\_\phi(z|\tau^0) \\\\ {\tau^{1:N}\sim q(\tau^{1:N}| \tau^0)}}}
> \sum\_{i=1}^{N}
> \log \frac{p\_\theta(\tau^{i-1}| \tau^i, z)}{p\_{\theta^{\text{old}}}(\tau^{i-1}| \tau^{i}, z)}
> \right) \right]  \\\\
> &= \mathbb{E}\_{\tau^{0} \sim p\_{E}}
> \left [ \log \sigma \left(
> \sum\_{i=1}^{N}
> \mathbb{E}\_{\substack{z\sim q_\phi(z|\tau^0) \\\\ {\tau^{1:N} \sim q(\tau^{1:N}| \tau^0)}}}
> \log \frac{p\_\theta(\tau^{i-1}| \tau^i, z)}{p\_{\theta^{\text{old}}}(\tau^{i-1}| \tau^{i},z)}
> \right) \right]  \\\\
> &= \mathbb{E}\_{\tau^{0} \sim p\_{E}}
> \left[ \log \sigma \left(
> \sum\_{i=1}^{N} \mathbb{E}\_{\substack{z\sim q\_\phi(z|\tau^0) \\\\ \tau^{i-1}, \tau^{i} \sim  q(\tau^i | \tau^0) q(\tau^{i-1}|\tau^i, \tau^0)}}
> \log \frac{p\_\theta(\tau^{i-1}| \tau^i, z)}{p\_{\theta^{\text{old}}}(\tau^{i-1}| \tau^{i},z)}
> \right) \right]  \\\\
> &\geq \mathbb{E}\_{\tau^{0} \sim p\_{E}}
> \left [ \mathbb{E}\_{\substack{i \sim \mathcal{U}(1, N),\\\\ z\sim q\_\phi(z|\tau^0), \tau^{i} \sim q(\tau^i|\tau^0)}}
> \log \sigma \left( {N} \mathbb{E}\_{\tau^{i-1} \sim q(\tau^{i-1}|\tau^i, \tau^0)}
> \log \frac{p\_\theta(\tau^{i-1}| \tau^i, z)}{p\_{\theta^{\text{old}}}(\tau^{i-1}| \tau^{i}, z)}
> \right) \right] (\text{concavity}) \\\\
> &= \mathbb{E}\_{\tau^{0} \sim p\_{E}}\left [
> \mathbb{E}\_{\substack{i\sim \mathcal{U}(1, N),\\\\ z\sim q\_\phi(z|\tau^0), \tau^{i} \sim q(\tau^i|\tau^0)}}
> \log \sigma
> \left( N \mathbb{E}\_{\tau^{i-1} \sim q(\tau^{i-1}|\tau^i, \tau^0)}
> \log \frac {q(\tau^{i-1}|\tau^i, \tau^0)}{p\_{\theta^{\text{old}}}(\tau^{i-1}| \tau^i,z)}
> -\log \frac {q(\tau^{i-1}|\tau^i, \tau^0)}{p\_\theta(\tau^{i-1}| \tau^{i},z)}\right) \right]
> \end{align*}$
>
> $\therefore \mathcal{L}\_{\text{DPAIL}}^{(1)}(\theta,\theta^{\text{old}},\tau^0, z) =
> \mathbb{E}\_{\tau^0, z, i, \epsilon}
> \\Big[ \log \sigma \\Big (N
> \\left (\\| \epsilon - \epsilon\_{\theta^{\text{old}}}(\tau^i,z,i)\\|^2
> \- \\| \epsilon - \epsilon\_\theta(\tau^i,z,i)\\|^2
> \\right )\\Big )\\Big],
> $
>
> Here, $z \sim q\_\phi(z|\tau^0)$, $\epsilon \sim \mathcal{N}(0, I)$, and $i \sim \mathcal{U}(1, N)$.
>
> Similarly, for given generative samples $\bar{\tau}\sim p\_{\theta^\{\text{old}}}$, we can also get $ \mathcal{L}\_{\text{DPAIL}}^{(2)}(\theta,\theta^{\text{old}},\bar{\tau}^0, z)$.
>
> The overall objective becomes:
>
> $\therefore
> \mathbb{E}\_{\substack{\tau^0 \sim p\_E, \\\\ z \sim q\_\phi(z|\tau^0)}}\\left[
> \mathcal{L}^{(1)}\_{\text{DPAIL}} (\theta, \theta^{\text{old}}, \tau^0,z) \\right]
> \+
> \mathbb{E}\_{\substack{\bar{\tau}^0 \sim p\_{\theta^{\text{old}}}, \\\\ z \sim q\_\phi(z|\bar{\tau}^0)}}
> \\left[ \mathcal{L}^{(2)}\_{\text{DPAIL}}(\theta, \theta^{\text{old}}, \bar{\tau}^0, z) \\right] +
> \text{MI}\_\phi(\bar{\tau}^0, z)
> $
>
> Where the last term encourages learning meaningful unsupervised representation $z \sim q\_\phi(z|\bar{\tau})$ by maximizing mutual information.

---

> > ### Comment · Reviewer_hTNL · 2025-08-05
> > **Replying to Rebuttal by Authors**
> >
> > Thank you for the authors’ detailed and well-structured rebuttal. The clarifications and additional analyses have addressed the majority of my concerns.  I encourage the authors to incorporate the clarifications and supporting results (e.g., sensitivity analysis, theoretical extensions) into the final version of the paper where appropriate, as they will enhance the overall clarity and completeness of the work.

---

### Official Review · Reviewer_MA5d · 2025-07-02

**Clarity:** 3
**Significance:** 3
**Originality:** 2
**Rating:** 4
**Confidence:** 4

**Summary:**

This paper introduces DPAIL, a novel adversarial imitation learning (AIL) framework that uses diffusion models as policies to better capture diverse expert behaviors in multimodal demonstration data. Building on the Adversarial Soft Advantage Fitting (ASAF) framework, DPAIL avoids policy optimization and instead uses a binary cross-entropy objective to distinguish between expert and generated trajectories. The method demonstrates good performance and robustness across various baselines, particularly as the number of behavior modes increases.

**Questions:**

Some symbols in the equation is not clearly defined. In Eq. (6), how is the equation satisfied? What is the definition of $\sigma$?

**Ethical Concerns:**

["NO or VERY MINOR ethics concerns only"]

**Final Justification:**

During the rebuttal, most of my concerns have been resolved. I think this is an interesting paper with good quality. Though there are some clarification issues, it could be addressed at the camera-ready phase.

**Limitations:**

Yes.

**Quality:**

3

**Strengths And Weaknesses:**

Pros:

1. This paper is well-written and easy to follow. The literature review is comprehensive and most related works are discussed.

2. This paper provides theoretical analysis to support the proposed objective function, which makes the method more convincing.

3. Empirical results from different aspects demonstrate the effectiveness of the proposed method, with several highly related works.

Cons:

1. This paper proposes a method that integrates the diffusion policy with the adversarial learning process. However, I do really think the authors' claim "without policy optimization" seems to be misleading. The first time I saw the title of this paper, I thought there was no policy update process within the adversarial imitation learning framework. After I read the whole paper, it seems that there still exists the optimization process of diffusion policy.

2. The formulation of discriminator in Eq. 5 is defined by the optimal result of the inner optimization loop in adversarial imitation learning. How does this formulation make sense? I understand this formulation is given by [5], however, as this work builds the foundation of this paper, the authors should make this point more clear.

3. I am also curious about the construction of multi-modal demonstration dataset. The authors claim that they use SAC to train the policy and select K expert policies during the training. This is more likely to be under the setting of imitation learning with imperfect demonstrations.

---

> ### Author Rebuttal · Authors · 2025-07-31
>
> **(Weakness 1) Clarifying the phrase “without policy optimization”**
>
> The phrase “without policy optimization” in our paper specifically refers to the absence of reward-driven policy gradient updates, which are commonly used in adversarial imitation learning frameworks.
> Our proposed method, DPAIL, does not perform RL optimization steps, such as policy gradients of value-based updates that maximize expected rewards.
>
> Instead, we train the diffusion policy using a lower bound objective derived from the discriminator, which does not incorporate any reward signal. As such, the optimization does not involve computing gradients of a reward-based objective with respect to the policy parameters. We will clarify this distinction in the paper.
>
> **(Weakness 2 & Question 1) Clarifying the discriminator formulation in Eq. (5) and Eq. (6)**
>
> In Eq. (5), we define the structured discriminator $D\_{\pi, {\pi'}}$ as
> $D\_{\pi, \pi'}(\tau) = {{p_{\pi'} (\tau) }\over{ p\_{\pi'} (\tau) + p\_\pi (\tau)}}$,
> which matches the form of the optimal discriminator in the inner maximization of eq. (4):
>
> $\max\_D \mathcal{L}(D, p_\pi)  = \max_D \mathbb{E}\_{\tau \sim p\_E}[\log D(\tau)] + \mathbb{E}\_{\tau \sim p_\pi} [\log (1 - D(\tau))]$.
>
> At optimality, the derivative of $\mathcal{L}(D, p\_\pi)$ with respect to $D$ should be zero for $\tau$:
>
> $\nabla\_D \mathcal{L}(D, p\_\pi) = \frac{p\_E(\tau) }{ D(\tau) } - \frac{p\_\pi(\tau)}{1-D(\tau)} = 0$.
>
> Rewriting this gives:
> $\frac{1-D(\tau)}{ D(\tau) }= \frac{p\_\pi(\tau) }{p\_E(\tau)}$.
>
> Therefore, the optimal solution is :
> $D^{*}\_{\pi, \pi\_E} = \frac{ p\_{\pi\_E(\tau)} }{ p\_{\pi\_E}(\tau) + p\_\pi (\tau) } = \frac{1}{1+ \frac{p\_\pi (\tau)}{p\_{\pi\_E(\tau)}}}=\frac{1}{1+ \exp(-\log\frac{p\_{\pi\_E}(\tau)}{p\_{\pi}(\tau)})} = \sigma \left(\log\frac{p\_{\pi\_E}(\tau)}{p\_{\pi}(\tau)} \right)$,
>
> where $\sigma(\cdot)$ is the sigmoid function.
>
> Thus, our definition in eq. (5) generalizes this form by replacing the expert policy $\pi\_E$ with a variable $\pi’$.
> We will clarify this in the paper.
>
> **(Weakness 3) How to construct multi-modal demonstration**
>
> To construct a multi-modal demonstration dataset, we use K sufficiently trained expert policies, each using a distinct and different reward function. For example, in AntGoal-v3, each reward function corresponds to the negative distance to a different target goal position from the current position.
> We collect demonstrations from all K experts and aggregate them into a single dataset. We consider all collected trajectories as perfect demonstrations.
> For detailed reward functions, please refer to Appendix B (Environment Details).

---

### Official Review · Reviewer_9YY9 · 2025-07-02

**Clarity:** 3
**Significance:** 3
**Originality:** 3
**Rating:** 4
**Confidence:** 4

**Summary:**

This paper introduces DPAIL, a novel framework for Adversarial Imitation Learning (AIL) that utilizes diffusion models as a policy function. The core innovation of DPAIL lies in its ability to learn diverse expert behaviors without requiring policy optimization, a common and computationally expensive step in traditional AIL methods. By building on the Adversarial Soft Advantage Fitting (ASAF) framework and introducing a tractable lower bound for training diffusion policies, DPAIL overcomes the limitations of mode collapse or instability when dealing with complex, varied demonstrations. The research demonstrates DPAIL's effectiveness in capturing diverse behaviors and maintaining robust performance across various continuous control and navigation tasks, even with limited demonstration data.

**Questions:**

How does the scaling factor $N$ affect optimization? Does clipping the denoising error itself suffices?

**Ethical Concerns:**

["NO or VERY MINOR ethics concerns only"]

**Final Justification:**

Thank you for the authors for the detailed responses. I feel my concerns have been addressed and my original rating is reasonable. I would maintain my score.

**Limitations:**

Yes

**Paper Formatting Concerns:**

No concern.

**Quality:**

3

**Strengths And Weaknesses:**

### Strength
+ The paper introduces the AIL framework using diffusion models as policies without requiring policy optimization based on the ASAF framework.
+ The paper derives a tight lower bound on the ASAF objective, enabling efficient training through the minimization-maximization approach.
+ The paper presents extensive experiments to demonstrate better performance than baselines works.

### Weakness
- Though the proposed approach does remove policy optimization for improved training efficiency, the proposed approach does come with higher computational cost and longer inference time.
- The scaling $N$ in eq. (9-10) is making the loss inside the sigmoid function large. This likely leads to the plateau region of the sigmoid function, hindering optimization.

---

> ### Author Rebuttal · Authors · 2025-07-31
>
> **(Weakness 1) High computation cost and longer inference time**
>
> As noted in the section on limitations, DPAIL incurs longer inference time due to the use of diffusion models as policies. This is a trade-off for their high expressiveness and ability to capture multi-modal behaviors, which are essential for our goal for modeling diverse expert demonstrations.
> There has been promising progress in accelerating diffusion model inference (e.g., through distillation or fast samplers), and integrating such techniques into our framework is a promising direction for improving its scalability and real-time applicability, which we leave as future work.
>
> **(Weakness 2 & Question 1) Scaling factor $N$ inside the sigmoid function**
>
> In our experiments, we set $N=50$, and we did not observe any numerical issues or optimization instability resulting from scaling factor $N$.
> Although $N$ increases the magnitude of the input to the sigmoid function in Eq. (9-10), this did not push it into the flat region of sigmoid in our experiments.
> This is because DPAIL resets the parameter via $\theta^{\text{old}} \leftarrow \theta$ at every iteration $k$, which causes the difference in noise prediction errors, the input of the sigmoid, to be zero at the start of each iteration.
> As a result, the input to the sigmoid remains within a manageable range. Specifically, for N = 50, the value typically lies within [0.0~8.0], where the log-sigmoid still provides meaningful gradients.
>
> For larger scaling factors (e.g N $\geq 100$), we investigated whether clipping the input to the log-simoid function would be necessary for the optimization. To assess this, we conducted additional experiments with N=100, both with and without clipping the input (e.g to a maximum value of 9).
> | task      | HalfCheetah | Walker | Ant | AntGoal | maze2d-medium | maze2d-large |
> |-|-|-|-|-|-|-|
> |       clip | 1.02 | 0.79 | 0.53 | 0.69 | 1.00 | 1.00 |
> | wo clip | 1.01 | 0.81 | 0.57 | 0.67 | 1.00 | 1.00 |
>
> In practice, we observed no significant difference in performance between them, likely because the input to the log-sigmoid resets to near-zero at the start of each iteration. However, the clipping may still be beneficial when using very large values of N for the effective optimization.

---

> > ### Comment · Reviewer_9YY9 · 2025-08-05
> >
> > Thank you for the authors for the detailed responses. I feel my concerns have been addressed and my original rating is reasonable. I would maintain my score.

---

### Official Review · Reviewer_96w4 · 2025-07-03

**Clarity:** 3
**Significance:** 2
**Originality:** 2
**Rating:** 4
**Confidence:** 3

**Summary:**

The paper studies bridging diffusion-based and adversarial-based imitation learning strategies. The key technical contribution lies in deriving a novel learning objective which integrates the adversarial soft advantage fitting (ASAF) objective into the diffusion model optimization. The proposed approach is tested under MuJoCo and Maze2D environments, justifying that the proposed approach brings the benefits of diffusion and adversarial approaches and outperforms existing imitation learning baselines.

**Questions:**

Please feel free to response to the issues listed in the weaknesses part.

**Ethical Concerns:**

["NO or VERY MINOR ethics concerns only"]

**Final Justification:**

Overall, I think this a technical sound paper which would be of interests in the imitation learning area. The author responses address my concerns in the reviews, thus I increase my score.

**Limitations:**

yes

**Quality:**

3

**Strengths And Weaknesses:**

- Strengths:

1. The proposed approach is based on sound derivation of variational lower bound.

2. The performance gain shown from the experiments is solid.

- Weaknesses:

1. The motivation needs further clarification. The paper does not provide clear explanations on why bridging diffusion and adversarial approaches is important. Does this help to address some specific open problem in imitation learning, or just bring the benefit of both approaches together? Answering this question is important for justifying the importance and value of the work.

2. The paper does not clearly explain the key technical challenge that the proposed approach tackles. Intuitively, a straightforward approach to bridging diffusion and adversarial learning is to utilize diffusion model as the policy, and directly add both diffusion and adversarial training objectives together to form a single objective. Why we need an alternative method derived in the paper? Justifying this would be important to evaluate the technical contribution in the paper.

3. Even though the results shown in the experiments verify the performance gain of the proposed approach, the benchmark tasks are relatively standard ones, which do not sufficiently show that the proposed approach can address some significantly challenging tasks that previous approaches fail to solve.

- Overall justification:

Overall, even though I think the paper is relatively technically sound, the weaknesses discussed above make me feel that the novelty and contributions of the paper are not sufficiently justified.

---

> ### Author Rebuttal · Authors · 2025-07-31
>
> **(Weakness 1) Motivation**
>
> In real-world settings, expert demonstrations are often limited in quantity and exhibit multi-modality. For example, there may be multiple distinct but valid strategies to solve the same task. Diffusion models are particularly well-suited to capture such multi-modal behavior due to their strong generative modeling capabilities. However, when only a few expert demonstrations are available, purely supervised learning on these data can lead to overfitting even for diffusion models.
> To address this, we bridge diffusion modeling with adversarial learning by leveraging diffusion-generated samples as negative examples. This enables the model to contrast expert data against generated ones, thereby promoting better alignment with expert distribution[1,2] and enhancing robustness[3] and generalization[4,5] of the learned policy.
>
> [1] Self-Play Fine-Tuning Converts Weak Language Models to Strong Language Models, Chen et al.,ICML 2024.
>
> [2] Adversarial Soft Advantage Fitting: Imitation Learning without Policy Optimization, Barde et al., NeurIPS 2020.
>
> [3] On the Convergence and Robustness of Adversarial Training, Wang et al., ICML 2019.
>
> [4] Adversarial Training Helps Transfer Learning via Better Representations, Deng et al., NeurIPS 2021.
>
> [5] Improved OOD Generalization via Adversarial Training and Pre-training, Yi et al., ICML 2021.
>
>  **(Weakness 2) Necessity of an alternative approach for diffusion policies in AIL**
>
> AIL methods typically involve training a discriminator and a generator in a minimax objective, which often leads to training instability. In particular, when the discriminator becomes too accurate, it can cause vanishing gradients for the generator, hindering the learning process.
> When applying such adversarial training to diffusion policies, these issues are further exacerbated. Specifically, computing policy gradient based on reward signals requires backpropagation through the entire diffusion steps, resulting in substantial computational cost and additional instability [6,7].
> To address these challenges, several recent works have been proposed. For instance, [6] introduces a technique that gradually increases the number of trainable diffusion steps, while [7] proposes a reverse sampling method that estimates the mapping from noise to clean data to stabilize training.
> However, these techniques still require careful scheduling and additional sampling, making the training process complex.
> In contrast, our proposed method, DPAIL, builds upon the ASAF framework, which allows us to sidestep these difficulties. By eliminating both the explicit discriminator and policy gradient-based optimization, DPAIL enables stable and efficient training of diffusion models, allowing them to be effectively applied to multi-modal generation.
>
> [6] Towards Better Alignment: Training Diffusion Models with Reinforcement Learning Against Sparse Rewards, Hu et al., CVPR, 2025.
>
> [7] Efficient Online Reinforcement Learning for Diffusion Policy, Ma et al., ICML, 2025.
>
> **(Weakness 3) Evaluation on challenging manipulation task**
>
> We conducted additional experiments on two challenging environments: FrankaKitchen and adroit, which go beyond standard benchmarks and better reflect the complexity of real-world tasks.
> The FrankaKitchen environment involves goal completion tasks that require the agent to perform household activities, such as opening a microwave or turning on a light switch.
> For experiments, we collected a mix of 5 trajectories per task for 4 tasks: ‘microwave’, ‘kettle’, ‘light switch’, and ‘kettle’.
> In addition, we evaluate DPAIL on the adroit pen manipulation task, which requires a dexterous robot hand to control and orient a pen. This task is particularly challenging due to its high-dimensional and contact-rich nature. We used 25 human demonstrations collected from [8].
> The normalized returns are shown below:
> | |BC|Diffusion|GAIL|DiffAIL|DRAIL|InfoGAIL|ASAF|DPAIL|
> |-|-|-|-|-|-|-|-|-|
> |franka-kitchen-v0 |0.09 (0.12)|0.52 (0.05)|0.32 (0.02)|0.43 (0.13)|0.56 (0.07)|0.07 (0.08)|0.07 (0.11)|**0.69** (0.10)|
> |ardoit-pen-v0 | 0.20 (0.19) | 0.89 (0.06) | 0.01 (0.00) | 0.01 (0.02) | 0.01 (0.04) | 0.00 (0.01) | 0.40 (0.13) | **1.21** (0.19) |
>
> These results demonstrate that DPAIL consistently outperforms baseline methods, validating its effectiveness in realistic scenarios.
>
> [8] D4RL: Datasets for Deep Data-Driven Reinforcement Learning, Fu et al., arxiv.

---

> > ### Comment · Reviewer_96w4 · 2025-08-05
> > **Thanks for the responses**
> >
> > Thanks for the responses. I feel that all my initial concerns are sufficiently addressed. I would raise my score accordingly. Meanwhile, I find the discussions and additional experiments quite useful to strengthen the clarity of the paper. I suggest including them in the revision.

---

### Author Response · Authors · 2025-08-04
**Thank you for your Feedback**

We sincerely appreciate the reviewer’s thoughtful comments and constructive suggestions. During the rebuttal period, we have made substantial efforts to address the questions:
- Clarified the motivation and technical justification of DPAIL. (Reviewer 96w4)
- Conducted additional experiments on more complex and realistic benchmarks, including Frankakitchen and Adroit environments. (Reviewer 96w4)
- Provided further analysis on the sensitivity of hyperparameters. (Reviewer hTNL)
- Extended DPAIL lower bound to incorporate latent variables for conditional modeling. (Reviewer hTNL)
- Clarified the discriminator formulation Eq.(5) and Eq.(6). (Reviewer MA5d)
- Clarified the usage of the phrase “without policy optimization”. (Reviewer MA5d)
- Analyzed the effect of the scaling factor $N$ inside the sigmoid function. (Reviewer 9YY9)

We hope these additions help clarify our contributions and strengthen the evaluations. We would greatly appreciate any further thoughts or feedback on our responses.

Thank you again for your time and positive consideration.

---

### Decision · Program_Chairs · 2025-09-17

**Decision:**

Accept (poster)

**Comment:**

The paper studies adversarial imitation learning with diffusion policies. To avoid the high computational cost and instability of RL-based policy optimization, the paper adopts the Adversarial Soft Advantage Fitting framework. On the upside:
- the reviewers acknowledge the principled derivation of the optimization objective
- the reviewers also acknowledged the solid experiments
- the reviewers addressed hyperparameter tuning fairness and sensitivity in the rebuttal.

on the downside:
- the reviewers are still concerned about the higher computational cost and longer inference time of the proposed approach
- the reviewers think that experiments remain focused on simulated environments; real-world or noisy/heterogeneous scenarios are not tested, leaving generalizability uncertain.

In summary, the paper is on the borderline.